# The molecular basis of phosphite and hypophosphite recognition by ABC-transporters

Claudine Bisson [1,3], Nathan B.P. Adams [1], Ben Stevenson[1], Amanda A. Brindley[1], Despo Polyviou[2], Thomas S. Bibby[2], Patrick J. Baker [1], C. Neil Hunter [1] & Andrew Hitchcock [1]

Inorganic phosphate is the major bioavailable form of the essential nutrient phosphorus. However, the concentration of phosphate in most natural habitats is low enough to limit microbial growth. Under phosphate-depleted conditions some bacteria utilise phosphite and hypophosphite as alternative sources of phosphorus, but the molecular basis of reduced phosphorus acquisition from the environment is not fully understood. Here, we present crystal structures and ligand binding affinities of periplasmic binding proteins from bacterial phosphite and hypophosphite ATP-binding cassette transporters. We reveal that phosphite and hypophosphite specificity results from a combination of steric selection and the presence of a P-H...π interaction between the ligand and a conserved aromatic residue in the ligand-binding pocket. The characterisation of high affinity and specific transporters has implications for the marine phosphorus redox cycle, and might aid the use of phosphite as an alternative phosphorus source in biotechnological, industrial and agricultural applications.

[1] Department of Molecular Biology and Biotechnology, University of Sheffield, Firth Court, Western Bank, Sheffield S10 2TN, UK. [2] Ocean and Earth Sciences, National Oceanography Centre Southampton, University of Southampton, Waterfront Campus, European Way, Southampton SO14 3ZH, UK. [3]Present address: Department of Biological Sciences, Crystallography ISMB, Birkbeck, University of London, Malet Street, London WC1E 7HX, UK. Claudine Bisson and Nathan B.P. Adams contributed equally to this work. Correspondence and requests for materials should be addressed to A.H. (email: a.hitchcock@sheffield.ac.uk)

Phosphorus (P), a component of nucleic acids, nucleotide cofactors and phospholipids, is essential to cell structure, metabolism and signalling. In nature it is mainly present in its most oxidised state (P valence = +5) predominantly as inorganic phosphate, which is the only form that cells can utilise directly for incorporation into biological molecules[1–3]. However, phosphate availability can vary significantly in natural environments and despite the presence of high-affinity phosphate uptake systems, in many ecosystems is sufficiently depleted to limit microbial growth. For example, phosphate concentrations in the oligotrophic surface waters of the vast ocean gyres can be extremely low, influencing the biogeochemistry and productivity of marine microorganisms[4, 5]. When phosphate is unavailable, some bacteria have evolved mechanisms to metabolise reduced phosphorus compounds to acquire P[2, 3, 6–10]. ATP-binding cassette (ABC) transporters for the active uptake of these compounds are encoded in the genomes of a range of microorganisms, with the transporter genes usually arranged in an operon with genes encoding enzymes that catalyse oxidation of the imported species to phosphate[11, 12] (Supplementary Fig. 1). Phosphonates (P valence = +3) are a diverse range of organic C-P bond containing compounds ubiquitous in natural habitats[9]. There are multiple mechanisms for their breakdown, the most common of which is the carbon-phosphorus (C-P) lyase, which converts phosphonates to a hydrocarbon and inorganic phosphate[6, 8, 13]. The crystal structure and ligand binding properties of the periplasmic binding protein (PBP) subunit from an *Escherichia coli* C-P lyase linked organophosphonate transporter have been reported previously[14, 15]; here we focus on the uptake of the alternative inorganic reduced phosphorus compounds phosphite ($HPO_3H^-/HPO_3^{2-}$; P valence = +3) and hypophosphite ($H_2PO_2^-$; P valence = +1).

The concentration of phosphite and hypophosphite in many natural environments is unknown, but they have been detected in up to micromolar amounts in some aquatic systems, where they can account for up to a third of the total dissolved P[16, 17]. The major sources of phosphite and hypophosphite are likely anthropogenic in origin (e.g. fungicides in agriculture), but they may also be derived from geothermal and biological processes[16, 17]. Many bacterial species are capable of utilising phosphite as a P source by oxidising it to phosphate (assimilatory phosphite oxidation)[12, 17–20]. The deltaproteobacteria *Desulfotignum phosphitoxidans* can also use phosphite as an electron donor and energy source by coupling its oxidation to the reduction of sulphate, $CO_2$ or nitrate (dissimilatory phosphite oxidation)[21, 22]. Phosphite is highly water-soluble and kinetically stable, and its oxidation by the NAD:phosphite oxidoreductase/phosphite dehydrogenase (PtxD) is highly exergonic (the reduction potential of phosphate/phosphite couple at pH 7 = −650 mV resulting in a $\Delta G = 63.7$ kJ mol$^{-1}$) and regenerates reductant in the form of NAD(P)H[2, 23]. Phosphite is also non-toxic and

inexpensive and, when provided as the sole source of P, it allows PtxD to be used as a selective marker, which has resulted in its use in numerous biotechnological and agricultural applications[24–29]. Hypophosphite can also be used as a sole P source by some microorganisms[18, 19, 30, 31]. The hypophosphite/2-oxoglutarate dioxygenase (HtxA) oxidises hypophosphite to phosphite (reduction potential at pH 7 = −740 mV) coupled to the oxidative decarboxylation of 2-oxoglutarate to succinate[32]; the phosphite produced is subsequently further oxidised to phosphate by PtxD.

The energetically favourable oxidation of phosphite and hypophosphite, yielding phosphate and NADPH, makes them attractive P sources to microorganisms under phosphate-limited conditions; however, the potentially low environmental concentration of these compounds necessitates the need for high-affinity transporters. While the genetic and biochemical basis of phosphite and hypophosphite utilisation and the PtxD and HtxA enzymes have been characterised previously[18, 23, 32, 33], the mechanism of high-affinity ligand binding by the cognate ABC transporters is not well studied. The use of non-physiologically high concentrations (up to 1 mM) of P ligands in previous in vivo studies has hindered clear interpretation of the physiologically relevant specificity of reduced P compound transporters[18, 19, 30, 34]. Here we elucidate the molecular basis of high-affinity phosphite and hypophosphite acquisition by determining the ligand-binding affinity and structure of the PBPs of phosphite and hypophosphite ABC transporters from environmental microorganisms. Our results reveal how a ligand-protein P-H…π interaction coupled with steric selection forms the basis of ligand specificity, explaining how these transporters allow bacteria to scavenge these alternative sources of P with ultra-high affinity.

## Results

**Ligand specificity of recombinant binding proteins**. We over-produced and purified PBPs of three putative phosphite transporters; PtxB from the globally important marine diazotroph *Trichodesmium erythraeum* IMS101 (Te_PtxB)[20], PhnD from the oceanic picocyanobacteria *Prochlorococcus marinus* MIT9301 (Pm_PhnD)[35] and PtxB from the soil bacterium *Pseudomonas stutzeri* WM88 (Ps_PtxB)[18], as well as a PBP from a putative hypophosphite transporter, HtxB, also from *P. stutzeri* WM88 (HtxB)[18] (Supplementary Fig. 2). The binding affinity of each protein for a range of P ligands was measured by microscale thermophoresis (MST)[36], using a RED-tris-NTA label (Nanotemper Technologies GmbH) via the C-terminal His-tag (Table 1; Supplementary Fig. 3). Dyes conjugated to the tris-NTA moiety provide a stable and very selective method to non-covalently fluorescently label proteins[37]. Dissociation constants ($K_d$) are derived from three independent MST experiments and errors

**Table 1 Microscale thermophoresis determined dissociation constants ($K_d$) of Te_PtxB, Ps_PtxB, Pm_PhnD and HtxB for selected P-ligands**

| Ligand | $K_d$ (μM) | | | |
|---|---|---|---|---|
| | Te_PtxB | Ps_PtxB | Pm_PhnD | HtxB |
| Phosphate | 46.80 ± 3.45 | 159.5 ± 36.0 | 182.2 ± 26.3 | NBD |
| Phosphite | 0.17 ± 0.07 | 0.24 ± 0.03 | 0.051 ± 0.004 | >10 × 10³ |
| Methylphosphonate | 30.63 ± 2.48 | 40.64 ± 5.70 | 108.8 ± 8.3 | NBD |
| Ethylphosphonate | >60 × 10³ | — | 111.9 ± 8.7 | — |
| 2-Aminoethylphosphonate | NBD | NBD | NBD | — |
| Hypophosphite | 2.21 ± 0.16 | — | — | 0.56 ± 0.11 |

$K_d$ values presented are derived from fitting three independent experiments to Eq. (1) and errors reported as ± the estimated standard deviation of the $K_d$. NBD indicates no binding was detected up to a ligand concentration of 10 mM; — indicates $K_d$ not measured

**Table 2 Data collection and refinement statistics part 1**

|  | Te_PtxB/phosphite (PDB: 5JVB) | Te_PtxB/MPn (PDB:5LQ1) | Pm_PhnD/phosphite (PDB:5LQ5) | Pm_PhnD /MPn (PDB:5LQ8) | Ps_PtxB apo (PDB: 5O2K) |
|---|---|---|---|---|---|
| *Data collection* |  |  |  |  |  |
| Wavelength (Å) | 0.96861 | 0.97629 | 0.97951 | 0.97951 | 0.97951 |
| Beamline | i24 | i03 | i24 | i24 | i04 |
| Resolution (Å) | 47.96–1.95 (2.00–1.95) | 65.05–1.41 (1.43–1.41) | 44.27–1.46 (1.49–1.46) | 44.0–1.52 (1.55–1.52) | 79.75–2.1 (2.14–2.1) |
| Space group | $P2_1$ | P1 | $P2_1$ | $P2_1$ | $P2_1$ |
| Unit cell ($a$, $b$, $c$, $\alpha$, $\beta$, $\gamma$) | 54.34, 69.31, 66.53, 90, 93.07, 90 | 39.48, 53.70, 66.39, 84.82, 93.07, 69.74 | 46.18, 57.41, 54.57, 90, 107.1, 90 | 45.86, 57.08, 54.17, 90, 106.56, 90 | 87.947, 136.640, 90.844, 90, 115.022, 90 |
| Total reflections[a] | 125,055 (9783) | 302,564 (10,007) | 162,732 (4664) | 88,958 (3373) | 762,654 (37,446) |
| Unique reflections[a] | 35,004 (2628) | 91,920 (3601) | 46,137 (1859) | 40,089 (1854) | 113,030 (5565) |
| Multiplicity[a] | 3.6 (3.7) | 3.3 (2.8) | 3.5 (2.5) | 2.2 (1.8) | 6.7 (6.7) |
| Completeness (%)[a] | 97.2 (98.9) | 95.2 (75.4) | 97.3 (78.4) | 97 (90.4) | 99.9 (98.7) |
| Mean I/σ (I)[a] | 7.3 (1.3) | 9.1 (1.1) | 15.8 (1.1) | 16.2 (1.2) | 6.3 (1.2) |
| $CC_{half}$[a] | 0.994 (0.554) | 0.999 (0.48) | 0.999 (0.569) | 0.997 (0.679) | 0.992 (0.535) |
| $R_{merge}$[a,b] | 0.081 (0.825) | 0.044 (0.813) | 0.035 (0.934) | 0.021 (0.434) | 0.174 (1.408) |
| $R_{pim}$[a,c] | 0.074 (0.725) | 0.032 (0.664) | 0.021 (0.670) | 0.021 (0.431) | 0.078 (0.625) |
| *Refinement* |  |  |  |  |  |
| $R_{factor}$/$R_{free}$ | 0.215/0.267 | 0.180/0.230 | 0.146/0.180 | 0.158/0.207 | 0.236/0.283 |
| RMSD bonds | 0.0106 | 0.0125 | 0.0127 | 0.0135 | 0.0121 |
| RMSD angles | 1.48 | 1.51 | 1.47 | 1.52 | 1.53 |
| No. of non-H atoms |  |  |  |  |  |
| Protein | 3925 | 4016 | 2225 | 2192 | 12,071 |
| Ligands | 8 | 18 | 4 | 10 | 0 |
| Water | 41 | 213 | 129 | 91 | 300 |
| Protein residues | 504 (2 chains) | 514 (2 chains) | 273 (1 chain) | 271 (1 chain) | 1562 (6 chains) |
| Average B-factors |  |  |  |  |  |
| Main chain/side chain | 40.5/45.8 | 19.1/28.6 | 25.6/38.0 | 29.7/40.2 | 34.9/38.6 |
| Ligands/solvent | 26.9/36.6 | 20.8/26.8 | 16.4/33.8 | 25.2/34.2 | - /26.7 |
| Ramachandran favoured/allowed (%) | 97.42/100 | 97.07/100 | 98.81/100 | 97.78/100 | 90.56/100 |
| Molprobity score | 1.17 (100th percentile $N$ = 13,349, 1.95 ± 0.25 Å) | 1.01 (99th percentile $N$ = 3343, 1.41 ± 0.25 Å) | 1.08 (99th percentile $N$ = 4372, 1.46 ± 0.25 Å) | 1.05 (99th percentile $N$ = 4870, 1.52 ± 0.25 Å) | 1.47 (98th percentile $N$ = 11,758, 2.1 ± 0.25 Å) |

MPn methylphosphonate

[a]Values in parenthesis are for data in the high-resolution shell

[b]$R_{merge} = \Sigma_{hkl} \Sigma_i \mid I_i - I_m \mid / \Sigma_{hkl} \Sigma_i I_i$

[c] $R_{pim} = \Sigma_{hkl} \sqrt{1/n - 1} \Sigma_{i=1} \mid I_i - I_m \mid / \Sigma_{hkl} \Sigma_i I_i$, where $I_i$ and $I_m$ are the observed intensity and mean intensity of related reflections, respectively

reported as ± the estimated standard deviation of the $K_d$. Te_PtxB, Pm_PhnD and Ps_PtxB all bound phosphite with a nanomolar $K_d$ in the range of ~50–240 nM (Table 1; Supplementary Fig. 3). In contrast, all three proteins bound methylphosphonate (MPn) and phosphate less tightly, with dissociation constants between ~30–100 and ~50–180 μM, respectively (Table 1; Supplementary Fig. 3); these values are an order of magnitude higher than the environmental concentration of these P compounds[35]. None of the proteins showed any evidence of binding the environmentally abundant[15] 2-aminoethylphosphonate (2AEPn). As an independent verification of the binding affinities we performed isothermal titration calorimetry (ITC) with Te_PtxB with phosphite ($K_d = 289 \pm 64$ nM) and phosphate ($K_d = 15.9 \pm 0.36$ μM), which gave dissociation constants in good agreement with those determined by MST (Supplementary Fig. 4). Furthermore, our MST-derived binding affinities for Pm_PhnD were similar to those reported previously by Feingersch et al.[35]. The proposed hypophosphite transporter, HtxB, bound hypophosphite with high affinity ($K_d = 560 \pm 11$ nM), with some evidence of phosphite binding in the millimolar range but no binding detected for phosphate (Supplementary Fig. 5) or methylphosphonate up to a concentration of 10 mM (Table 1). These data show that at environmentally relevant concentrations Te_PtxB, Pm_PhnD and Ps_PtxB clearly favour phosphite as a ligand, and HtxB appears to be specific for hypophosphite.

**Phosphite binding involves a P-H…π interaction.** To determine the structural mechanism of phosphite selectivity we co-crystallised Te_PtxB and Pm_PhnD with phosphite resulting in high-resolution structures (1.95 and 1.46 Å respectively) of the closed, ligand-bound complexes (Table 2). Te_PtxB and Pm_PhnD are both typical type II PBPs[38], with two lobes separated by a hinge region surrounding a buried, enclosed central cavity that forms the ligand-binding pocket (Fig. 1a, b; Supplementary Figs. 6–9) (RMSD cα: ~1.7 Å). In the closed form, the amino acids that form the walls of the binding pocket pack tightly against the phosphite moiety and a single water molecule that is also buried in the binding pocket. Each oxygen atom of the phosphite contributes to an extensive hydrogen bonding network to the main chain and side chains of a cluster of conserved residues from lobe 2 (Te_PtxB; Y55, Y100, S130, T131, S132 and H160; Pm_PhnD; Y46, S126, T127, S128, H158 and D205, Figs. 1d, e and 2e, f). In Pm_PhnD, the interaction of the D205 carboxylate with the ligand suggests that, under the crystallisation conditions used (pH 8), mono-anionic ($HPO_3H^-$) rather than di-anionic ($HPO_3^{2-}$) phosphite is the predominantly bound form. In Te_PtxB, it is not possible to determine the protonation state of the phosphite from the pattern of hydrogen bonding, but given the pH of the crystallisation experiment (pH 4.2), it is also likely that the mono-anionic form of phosphite is bound (phosphite $pK_a$ 1.5, 6.7[39]). Both proteins contain a conserved histidine as one of the phosphite

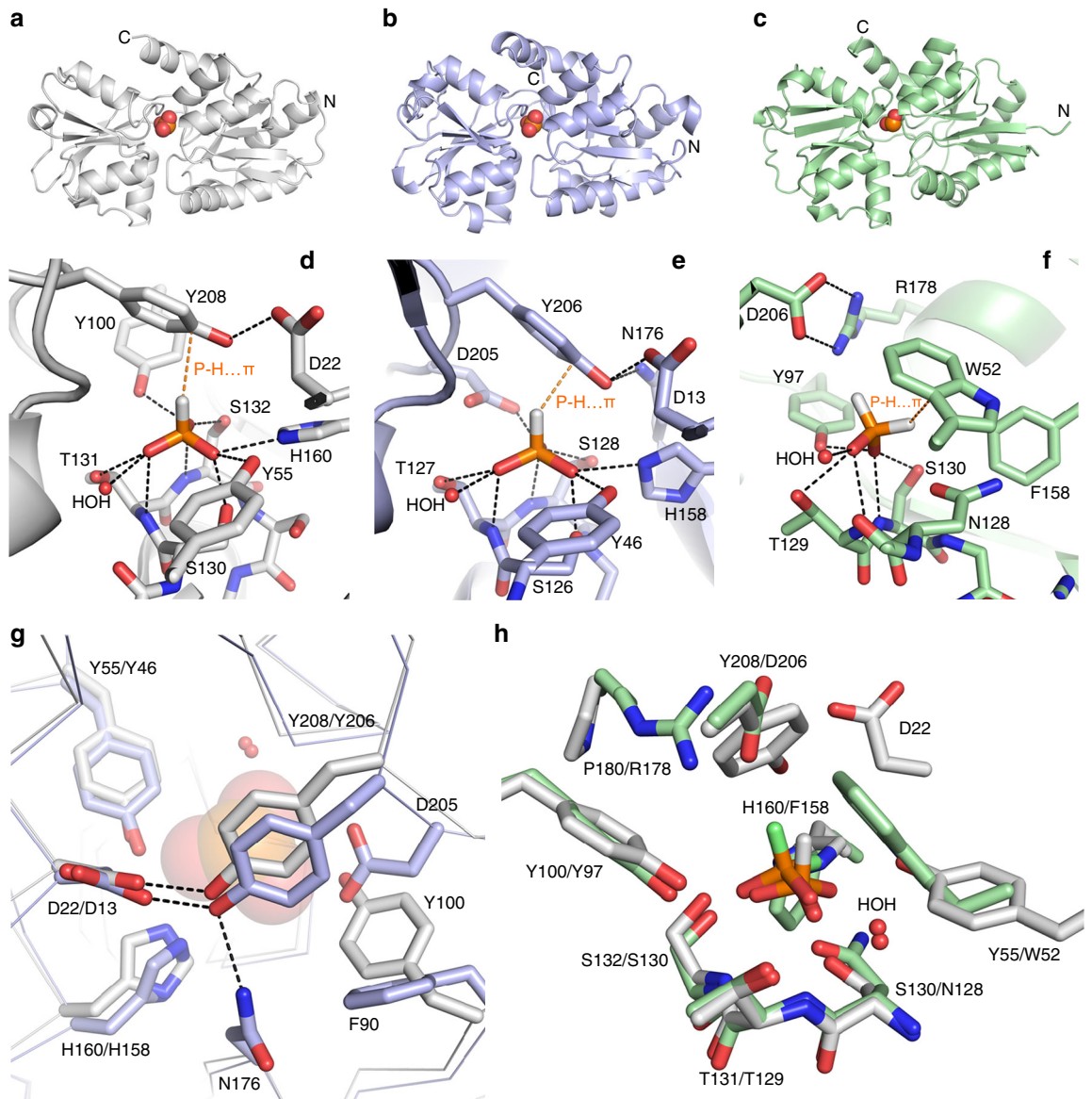

**Fig. 1** The crystal structures of Te_PtxB and Pm_PhnD in complex with phosphite and HtxB in complex with hypophosphite. **a–c** A cartoon representation of the overall fold of Te_PtxB (grey, **a**), Pm_PhnD (blue, **b**) and HtxB (green, **c**), with the ligand drawn as spheres and coloured in atom colours (N = blue, P = orange, O = red, S = yellow, H = white). The termini are labelled. See also domain topology diagram in Supplementary Fig. 6. **d–f** A detailed view of the ligand-binding pocket, highlighting the interactions made between the ligand and the protein for Te_PtxB (**d**), Pm_PhnD (**e**) and HtxB (**f**). The protein backbones are drawn as a cartoon, with the bound ligand and surrounding sidechains drawn as sticks, and coloured as in parts (**a–c**). The single buried water molecule (HOH) is drawn as a red sphere. Hydrogen bonds are drawn as dashed lines and the P-H...π interaction is drawn as an orange dashed line. **g** A superposition of Te_PtxB and Pm_PhnD, viewed from above the phosphite binding site, shows how exchanging Y100 in Te_PtxB for a phenylalanine (F90) in Pm_PhnD requires D205 to be recruited to complete the network of hydrogen bonds around the phosphite ligand. The phosphite is drawn as partially transparent spheres and the buried water molecule is indicated as a red sphere. **h** A comparison of the binding pockets of Te_PtxB and HtxB showing how the interactions around two of the P-oxygen binding sites are largely spatially conserved, but the third site is modified in HtxB (H160 to F158) to essentially block binding of a third oxygen moiety. The hydrogen atoms of the ligand are highlighted in green for HtxB and in white for Te_PtxB, with most of the main chain atoms omitted for clarity

ligands (Te_PtxB; H160 and Pm_PhnD; H158), which could help stabilise the negative charge on the phosphite. However, in Te_PtxB, one imidazole nitrogen atom acts as a hydrogen bond acceptor to a mainchain N-H (S178), suggesting that the histidine is neutral. In Pm_PhnD the equivalent interaction is made to a buried water molecule outside of the binding pocket and the hydrogen bonding network around this water molecule is not sufficient to determine the charge on the histidine.

In both structures, the hydrogen atom at the R1 position of the phosphite points towards lobe 1, packing against the face of the aromatic ring of a tyrosine residue (Te_PtxB; Y208, Pm_PhnD; Y206). This tyrosine sidechain forms a cap that is stabilised by intramolecular hydrogen bonds to residues on lobe 1 (Te_PtxB; D22, Pm_PhnD; D13 and N176), which in the closed conformation, contributes to the network of interactions that connect both lobes of the protein and bury the binding pocket from solvent (Fig. 2a, b, e, f). The distance between the R1 hydrogen and the plane of the aromatic ring of this tyrosine is ~2.6 Å, which is less than the sum of the expected van der Waals radii and, as the P-H bond also impinges upon the π system at an angle of ~140°, this

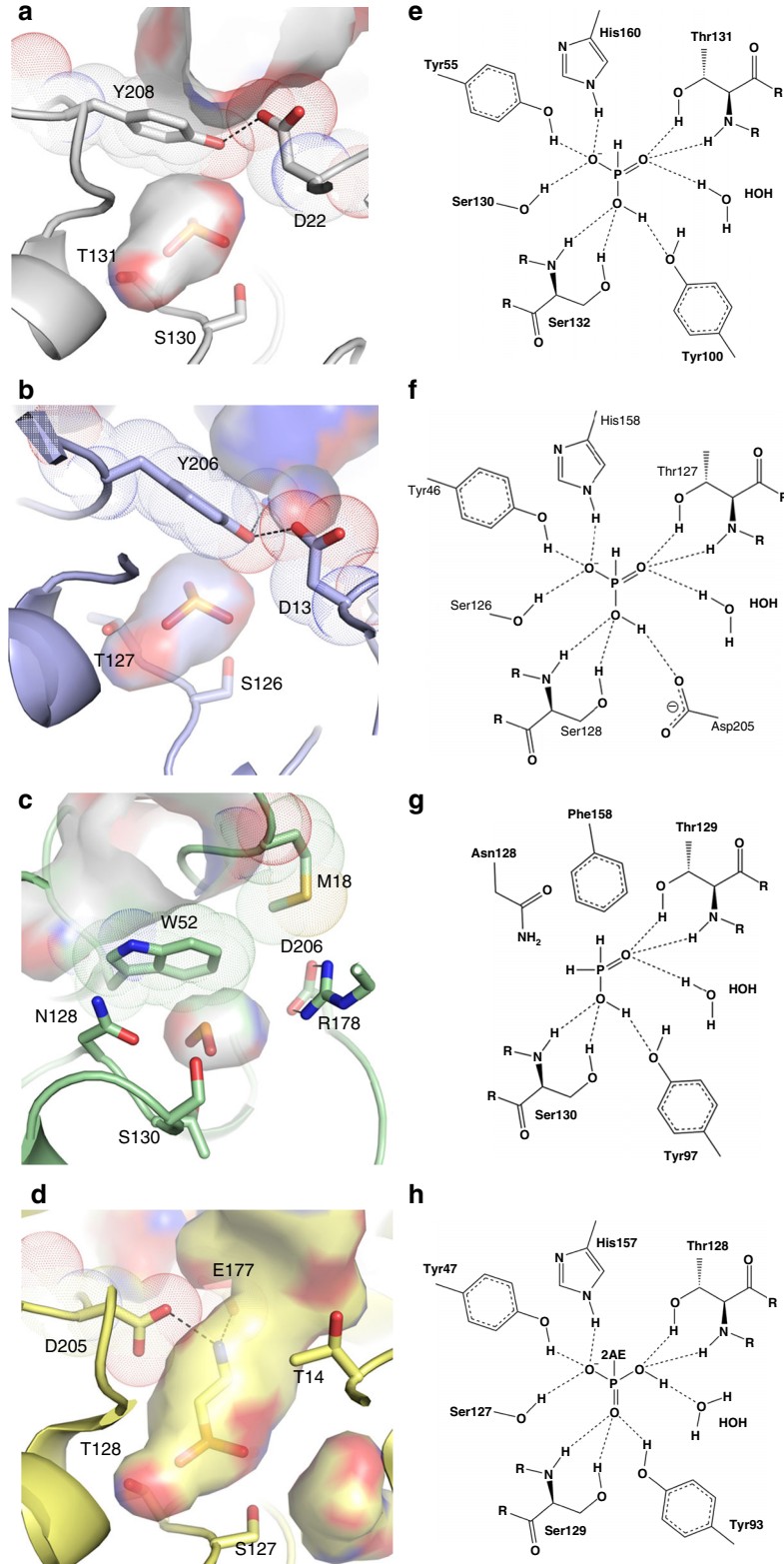

**Fig. 2** A detailed comparison of the spatial arrangement of the binding pocket and network of interactions around the ligand. The interior surface (partially transparent, atom colours) of the binding pocket and surrounding residues (sticks) in **a** Te_PtxB; **b** Pm_PhnD; **c** HtxB and **d** E. coli PhnD (PDB:3P7I). In each case, the capping residues are highlighted with a dotted surface that represents the van der Waals radii of the atoms. The external surface is also shown in each case, demonstrating how the binding pocket in reduced phosphorus binding PBPs (**a–c**) is buried from the external solvent and in each case is much smaller than that of the phosphonate binding PhnD from E. coli (**d**). The protein backbone is drawn as a cartoon, with sidechains drawn in sticks and hydrogen bonds between the capping residues indicated by black dashes. Each figure (**a–d**) is accompanied by a schematic (**e–h**) that displays the hydrogen-bonding network between the protein and the oxygen atoms of the ligand. The 2-aminoethyl group of 2AEPn is abbreviated to 2AE in **h**

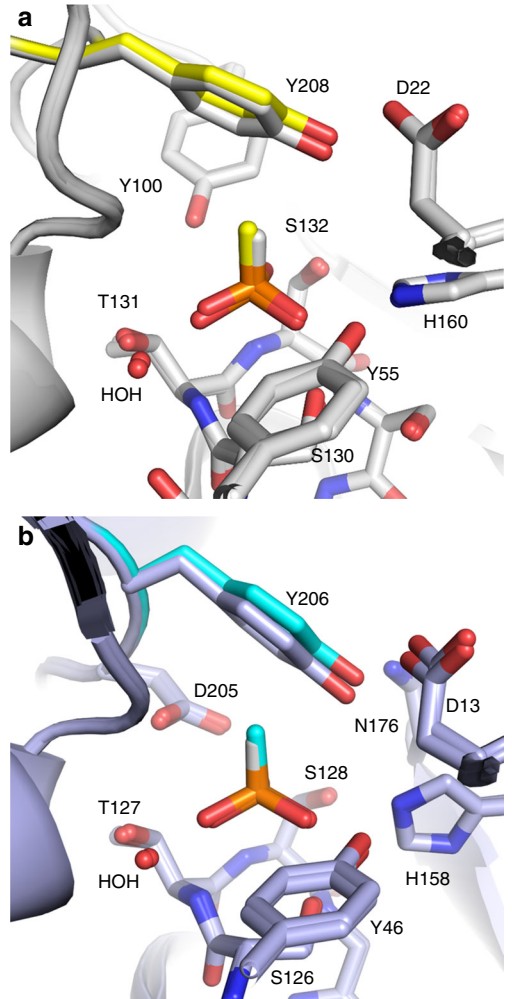

**Fig. 3** A comparison of phosphite and methylphosphonate binding in Te_PtxB and Pm_PhnD. **a** Superposition of the complexes of Te_PtxB (grey) with phosphite (white hydrogen) and methylphosphonate (MPn; yellow methyl) showing that the residues around the binding pocket are in consistent positions in each structure apart from the capping tyrosine which moves by ~0.5 Å in the MPn complex (Y208, yellow) in response to the larger van der Waals radii of the methyl group of the ligand. **b** Superposition of the phosphite (white hydrogen) and MPn (cyan methyl) complexes with Pm_PhnD (blue) showing a similar movement of the capping tyrosine (Y206, cyan)

suggests a P-H…π interaction occurs[40, 41] (Fig. 1d, e). To test the importance of the capping tyrosine for phosphite binding we generated mutants of Te_PtxB, exchanging the tyrosine for a phenylalanine or alanine; circular dichroism (CD) spectra showed that neither substitution changed the secondary structure of the protein (Supplementary Fig. 10). Both mutants show reduced capacity to bind phosphite, with ~2 and ~3 orders of magnitude weaker binding for the Y208A and Y208F mutants, respectively (Table 3; Supplementary Fig. 11). The reduced affinity of the Y208F mutant suggests that simply positioning a π electron system above the R1-hydrogen of the phosphite ligand is not enough to achieve stable phosphite binding. The tyrosine cap therefore appears to play a dual role, both providing a platform for the P-H…π interaction with phosphite and also stabilising the closed state by providing a hydrogen bond network between the two domains, centred on the tyrosine hydroxyl. We also tested the affinity of the Y208A and Y208F mutants for phosphate. Interestingly, the tyrosine to alanine mutation made little

difference to binding affinity, suggesting that phosphate may interact weakly with the protein in a partially open state, with few, if any interactions with the tyrosine side chain. The Y208F mutation essentially abolished phosphate binding, again showing the importance of the inter domain hydrogen bond network. Overall, this suggests that the tyrosine cap is a pre-requisite to achieve a fully closed state of the protein that engulfs the phosphite within the binding pocket and acts as a steric barrier for the binding of ligands with bulkier R groups, providing an explanation for the observed specificity of Te_PtxB and Pm_PhnD for phosphite.

Te_PtxB and Pm_PhnD have 2–4 orders of magnitude higher binding affinity for phosphite than phosphate and phosphonates (Table 1), but whilst attempts to crystallise complexes of either protein with ethylphosphonate (EPn), 2AEPn or phosphate were unsuccessful, structures of both proteins in complex with methylphosphonate (MPn) were obtained (Table 2; Fig. 3). The overall fold of the Te_PtxB and Pm_PhnD complexes with MPn were very similar to the comparative complexes with phosphite (RMSD cα: ~0.3 Å) and since both MPn complexes were determined at high resolution (1.41–1.52 Å), the electron density for the ligand could be confidently interpreted within the binding pocket, with the MPn refining unequivocally in only one orientation (Supplementary Figs. 12 and 13). The only difference in chemical structure between phosphite and MPn is an exchange of the R1 hydrogen in phosphite for a methyl group in MPn. Given this similarity, it was unsurprising that the three oxygen atoms and the phosphorus atom of the MPn bind in the same position as those of the phosphite, with the methyl group of the MPn pointing in the same direction as the R1 hydrogen of the phosphite, towards the capping tyrosine residue (Y208/Y206). To accommodate the methyl group of the MPn the tyrosine moves in both structures by ~0.5 Å, compared to the complexes with phosphite, resulting in an ~16% expansion in the volume of the binding pocket (Supplementary Table 1). Despite this movement, the methyl-carbon atom of the MPn lies ~3.0 Å from the plane of the tyrosine ring in both structures. This might suggest that a C-H…π interaction is present, however the relative geometry of the two partners is sub-optimal for this type of interaction[41]. Thermal ellipsoids were calculated from the anisotropic B-factors for the three structures that were better than 1.5 Å resolution (Te_PtxB and Pm_PhnD with MPn and Pm_PhnD with phosphite). These show an elongation of the ellipsoids for the tyrosine atoms in the MPn complexes, compared to those of in the phosphite complex, suggesting a degree of instability in the position of the tyrosine next to the methyl group of the MPn (Supplementary Fig. 14). Since Te_PtxB and Pm_PhnD have a significantly lower binding affinity for MPn than for phosphite (100- and 2000-fold, respectively), this suggests that the ~0.5 Å movement of the capping tyrosine may alter the network of hydrogen bonds that stabilise the closed, ligand-bound conformation. To explore this, we measured the binding affinity of the Y208A and Y208F mutants of Te_PtxB for MPn. The Ala substitution weakened the binding of MPn by a factor of 10, whilst the Phe mutation essentially abolished binding of MPn (Table 3 and Supplementary Fig. 11). This suggests that by swapping the bulky tyrosine for an alanine, the steric barrier provided by the Tyr is removed, providing space in the binding pocket for the methyl group of MPn. However, as the alanine is unable to form bridging hydrogen bonds to stabilise the closed conformation, the binding of MPn is weaker. CD shows that Y208F mutant is folded (Supplementary Fig. 10), however, we could not detect binding of MPn, suggesting that this amino-acid substitution blocks the closed ligand-bound complex from forming. Taken together, the exquisite selectivity of PtxB for phosphite can be explained not only by the favourable P-H…π

**Table 3 Microscale thermophoresis determined dissociation constants ($K_d$) of Te_PtxB and HtxB binding pocket mutants**

| Ligand | $K_d$ (μM) | | | | | | |
|---|---|---|---|---|---|---|---|
| | Te_PtxB | Te_PtxB Y208A | Te_PtxB Y208F | HtxB | HtxB W52A | HtxB W52F | HtxB W52Y |
| Phosphate | $46.80 \pm 3.45$ | $51.27 \pm 1.76$ | $>60 \times 10^3$ | NBD | — | — | — |
| Phosphite | $0.17 \pm 0.07$ | $134.5 \pm 6.4$ | $294.81 \pm 4.64$ | $>10 \times 10^3$ | — | — | — |
| Methylphosphonate | $30.63 \pm 2.48$ | $255.4 \pm 20.2$ | $>60 \times 10^3$ | NBD | — | — | — |
| Hypophosphite | $2.21 \pm 0.16$ | — | — | $0.56 \pm 0.11$ | NBD | $28.49 \pm 6.10$ | $147.54 \pm 14.48$ |

$K_d$ values presented are derived from fitting three independent experiments to Eq. (1) and errors reported as ± the estimated standard deviation of the $K_d$. NBD indicates no binding was detected up to a ligand concentration of 10 mM; - indicates $K_d$ not measured

**Table 4 Data collection and refinement statistics part 2**

| | Ps_PtxB/phosphite (PDB:5O2J) | Ps_PtxB/MPn (PDB:5O37) | HtxB/hypophosphite (PDB: 5ME4) | Pm_PtxB/phosphite (PDB: 5LV1) |
|---|---|---|---|---|
| *Data collection* | | | | |
| Wavelength (Å) | 0.92819 | 0.92819 | 0.97951 | 0.92819 |
| Beamline | i04 − 1 | i04 − 1 | i24 | i04 − 1 |
| Resolution (Å) | 33.25–1.52 (1.55–1.52) | 37.61–1.37 (1.39–1.37) | 22.81–1.52 (1.55–1.52) | 131.79–2.12 (2.16–2.12) |
| Space group | $P2_12_12$ | $P2_12_12$ | $P2_12_12_1$ | $P3_121$ |
| Unit cell ($a$, $b$, $c$, $\alpha$, $\beta$, $\gamma$) | 113.21, 39.03, 63.52, 90, 90, 90 | 112.8, 39.12, 63.3, 90, 90, 90 | 40.24, 55.21, 125.2, 90, 90, 90 | 152.175, 152.175, 67.915, 90, 90, 120 |
| Total reflections[a] | 548,602 (22,254) | 672,723 (21,736) | 303,161 (11,860) | 982,237 (39,470) |
| Unique reflections[a] | 44,234 (2169) | 59,851 (2940) | 42,787 (1986) | 51,504 (2522) |
| Multiplicity[a] | 12.4 (10.3) | 11.2 (7.4) | 7.1 (6.0) | 19.1 (15.7) |
| Completeness (%)[a] | 100 (100) | 99.9 (98.8) | 97.7 (91.3) | 100 (100) |
| Mean $I/\sigma$ ($I$)[a] | 11.9 (1.2) | 16.8 (1.2) | 9.8 (1.1) | 7.7 (1.4) |
| $CC_{half}$[a] | 0.999 (0.510) | 1 (0.485) | 0.998 (0.525) | 0.996 (0.648) |
| $R_{merge}$[a,b] | 0.127 (1.774) | 0.069 (1.463) | 0.117 (1.209) | 0.36 (2.82) |
| $R_{pim}$[a,c] | 0.039 (0.607) | 0.022 (0.615) | 0.05 (0.553) | 0.086 (0.733) |
| *Refinement* | | | | |
| $R_{factor}$/$R_{free}$ | 0.159/0.189 | 0.138/0.177 | 0.167/0.198 | 0.218/0.264 |
| RMSD bonds | 0.0131 | 0.015 | 0.0108 | 0.0112 |
| RMSD angles | 1.48 | 1.5 | 1.45 | 1.47 |
| No. of non-H atoms | | | | |
| Protein | 1998 | 2038 | 2034 | 6010 |
| Ligands | 8 | 5 | 6 | 12 |
| Water | 193 | 225 | 175 | 123 |
| Protein residues | 259 (1 chain) | 261 (1 chain) | 255 (1 chain) | 789 (3 chains) |
| Average B-factors | | | | |
| Main chain/side chain | 17.8/26.0 | 17.1/24.1 | 18.1/24.4 | 28.7/32.2 |
| Ligands/aolvent | 21.1/32.1 | 11.0/32.0 | 31.8/31.6 | 19.3/25.9 |
| Ramachandran favoured/ allowed (%) | 97.28/100 | 97.3/100 | 96.48/100 | 98.09/100 |
| Molprobity score | 0.73 (100th percentile $N = 4870$, $1.52 \pm 0.25$ Å) | 1.05 (99th percentile $N = 3026$, $1.37 \pm 0.25$ Å) | 1.18 (97th percentile $N = 4870$, $1.52 \pm 0.25$ Å) | 0.78 (100th percentile $N = 11300$, $2.12 \pm 0.25$ Å) |

MPn methylphosphonate
[a]Values in parenthesis are for data in the high-resolution shell
[b]$R_{merge} = \Sigma_{hkl}\Sigma_i \mid I_i - I_m \mid / \Sigma_{hkl}\Sigma_i I_i$
[c]$R_{pim} = \Sigma_{hkl} \sqrt{1/n - 1}\Sigma_{i=1} \mid I_i - I_m \mid / \Sigma_{hkl} \Sigma_i I_i$, where $I_i$ and $I_m$ are the observed intensity and mean intensity of related reflections, respectively

interaction between the phosphite and the protein, but also by a combination of a steric requirement for a hydrogen at the R1 position of the ligand and a strengthening network of hydrogen bonds around the binding pocket that stabilise the closed, phosphite-bound state.

**Ligand binding induces a large conformational change in PtxB.** Determining an apo-structure of a PBP from this family of ABC transporters is difficult due to the inherent flexibility of the hinge region between the two domains, and it has only been achieved previously with PhnD from *E. coli* (PDB:3S4U), which was locked in an open conformation by substituting H157 with an alanine[14]. In our study, numerous attempts to crystallise apo forms of Te_PtxB and Pm_PhnD were unsuccessful, but a 2.1 Å resolution structure of *Pseudomonas stutuzi* PtxB (Ps_PtxB) in an open conformation was determined (Table 2). We also obtained structures of phosphite and MPn complexed Ps_PtxB (Table 4; Fig. 4) and determined the associated ligand affinities (Table 1); both the structures and binding data were similar to the other phosphite-binding proteins discussed previously. Analysis of the conformational change between the open and closed, phosphite-bound structure of Ps_PtxB using DynDom[42] revealed that domain closure is induced by a ~60° rotation around an axis that lies between the two lobes (Fig. 4). This is facilitated by flexible loops that form the hinge region between the lobes (residues A85-T93 and Y200-Q202), as

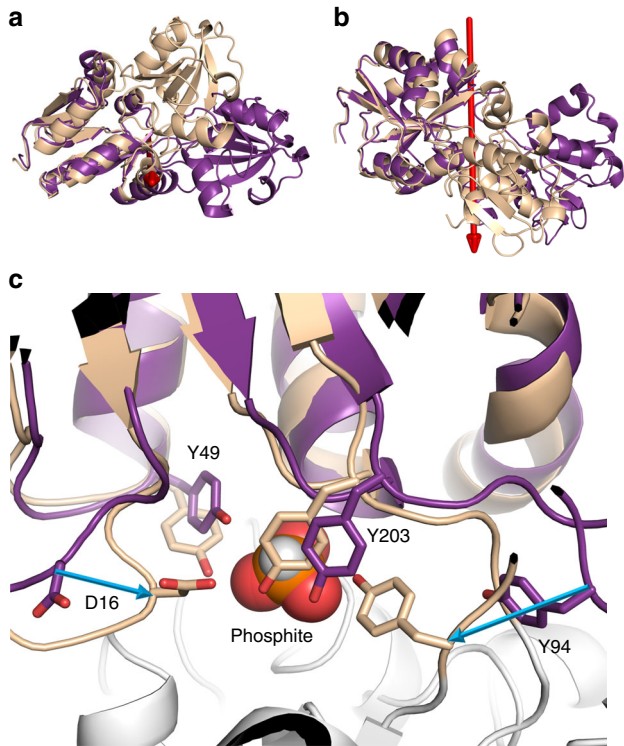

**Fig. 4** The conformational change between the open and closed structures of Ps_PtxB. **a, b** A comparison of the open (purple) and closed (beige) structure of Ps_PtxB, superimposed on lobe 2, showing the 60° rotation of one domain relative to the other around an axis that lies between the two domains (red arrow). **c** Superimposing the open and closed structures on lobe 2 (lobe 1 is shown in grey) shows that the spatial arrangement of Y49 and Y203 is conserved in both the open and closed states, whilst Y94 and D16, which both sit on flexible loop regions, undergo large conformational movements (blue arrows) of 7 and 4.5 Å (cα- cα), respectively. The phosphite is shown as spheres

well as residues from the extreme end of the C-terminal helix (G255-L258). These changes are similar to those observed between the open and closed structures of *E. coli* PhnD[14].

Superposition of the individual lobes of Ps_PtxB in the open and closed conformations shows that lobe 2 (residues 96–196, 0.35 Å RMSD cα) adopts a very similar structure, with the conserved STS-H motif (S124, T125, S126 and H155) spatially conserved in both conformations. In contrast, lobe 1 (residues 1–84 and 202–255, 1.55 Å RMSD cα) shows some differences between the open and ligand-bound states. The positions of Y94, which hydrogen bonds to one of the phosphite oxygen atoms in the closed structure, and D16, which hydrogen bonds to and stabilises the capping tyrosine in the closed structure, move ~7 Å and ~4.5 Å (cα-cα), respectively, on ligand binding. Y94 resides on the hinge region, the conformation of which alters substantially during domain closure, whilst D16 sits on a flexible loop that latches over the binding pocket in the closed conformation. In addition, the B-factors of the core residues in lobe 2 are much lower in the closed, ligand bound, conformation compared to those in the open structure, implying lower overall flexibility when phosphite is bound to the protein (Supplementary Fig. 15).

**HtxB binds hypophosphite using a similar P-H…π interaction**. The structure of HtxB in complex with hypophosphite (Table 4) shows that the protein adopts the same closed conformation as seen in the complex of Te_PtxB with phosphite (RMSD cα 1.86 Å) (Fig. 1c, f). The ligand-binding pocket is very similar, with

two of the hypophosphite oxygen atoms (O1 and O2) adopting the same position and making the same interaction as the equivalent oxygen atoms in the Te_PtxB/phosphite complex (Fig. 1f, h). However, H160, Y55 and S130, which all make interactions with O3 of the phosphite in Te_PtxB are replaced in HtxB with F158, W52 and N128, respectively. This has the effect of removing the hydrogen bonding potential around the O3 site, reducing the volume of the binding pocket by ~36 % (Fig. 2c, g; Supplementary Table 1) and providing steric selectivity for hypophosphite (two oxygen substituents) over phosphite (three oxygen substituents). Consistent with this, we observed only negligible levels of phosphite binding to HtxB in our MST assays (Table 1). Conversely, hypophosphite binds to Te_PtxB with low micromolar affinity ($K_d$ ~2 μM), indicating that presumably its smaller size can be accommodated within the Te_PtxB binding pocket, albeit with suboptimal interactions.

The changes in the ligand-binding pocket also extend to the capping residue, Y208, which in Te_PtxB provides the π system for the P-H…π interaction. In HtxB, a buried ion pair occurs in this region of the structure (D206 from lobe 1 and R178 from lobe 2), which connects the two lobes of the protein when it is in the closed conformation (Fig. 1f). Nevertheless, in HtxB the P-H…π interaction between the ligand and the protein is conserved by the presence of a tryptophan residue, W52 (Fig. 1h). The aromatic ring system of this tryptophan is positioned adjacent to the hydrogen at the R2 position of hypophosphite and is in the appropriate orientation for a P-H…π interaction to occur (~2.6 Å, 140°), in the same way as seen in the phosphite-binding proteins (Fig. 1f). Replacing this tryptophan with a phenylalanine or tyrosine substantially reduces the binding affinity of HtxB for hypophosphite by 50-fold and 250-fold, respectively (Table 3; Supplementary Fig. 11). The smaller size of both sidechains would clearly alter the packing within this region of the binding pocket and may not provide a large enough platform for the R2 hydrogen of the hypophosphite to pack against. The lower binding affinity in the tyrosine mutant may be due to its hydroxyl group interfering with Phe158 on the adjacent side of the binding pocket, possibly blocking closure of the protein around its substrate. Changing the tryptophan to an alanine results in the complete abolition of binding, suggesting the packing interactions between the tryptophan, nearby residues and the ligand are critical for high-affinity binding of hypophosphite. Therefore, we have demonstrated that in both phosphite and hypophosphite binding proteins, positioning the π electron system of an aromatic amino acid adjacent to the P-H atom of the ligand is a conserved molecular mechanism governing specificity of the PBPs for reduced phosphorus ligands.

**Structural comparisons with phosphonate binding PBPs**. A phylogenetic tree of members of the phosphonate/phosphite/phosphate periplasmic substrate binding protein family (KEGG orthology K02044; InterPro Family IPR005770; Pfam12974) shows they cluster into five main sub-groups based on primary sequence (Fig. 5a). These groups are: (1) C-P lyase-linked phosphonate transporters (PhnD); (2) phosphite dehydrogenase (PtxD)-linked phosphite transporters (PtxB); (3) the PhnD PBP of the marine PhnDCE phosphite transporter; (4) hypophosphite transporters (HtxB); and (5) transporters we have referred to as 'hybrid' as they share sequence features with both PtxB and HtxB. We have determined the structure of three proteins from group 2 (Te_PtxB, Pm_PtxB and Ps_PtxB) and proteins from groups 3 (Pm_PhnD) and 4 (HtxB). From structure-based sequence alignments of representative members of these groups it is clear that the residues imparting phosphite and hypophosphite specificity are strictly conserved (Fig. 5b). The only

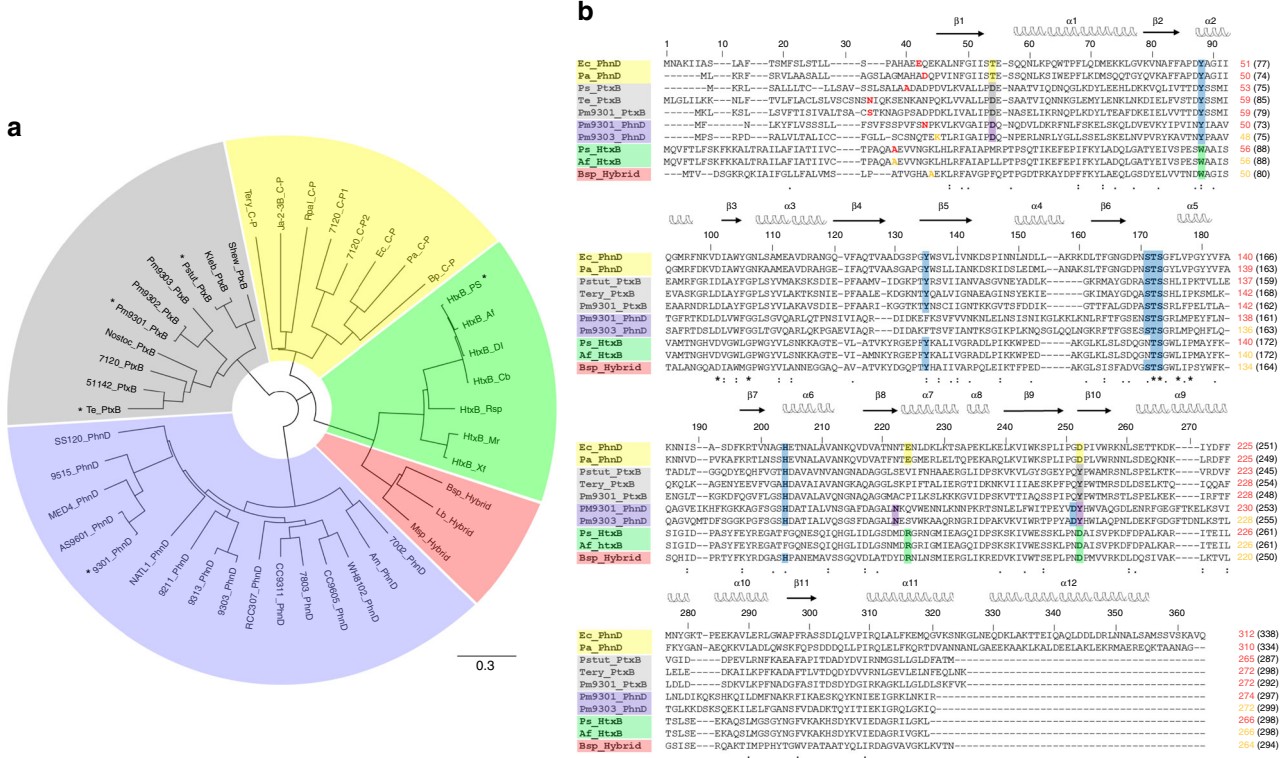

**Fig. 5** Protein phylogeny and sequence alignments of reduced phosphorus compound PBPs. **a** Unrooted phylogenetic tree of reduced P compound transporter PBP amino-acid sequences. Phosphite binding PtxB (grey) and PhnD (purple) homologues cluster separately from each other and from C-P lyase-linked phosphonate binding PhnDs (yellow), HtxB homologues (green) and a putative 'hybrid' class of transporters (red). Proteins characterised in this study are marked with an asterisk and the scale bar represents the number of substitutions per site. **b** Sequence alignment of reduced phosphorus compound binding PBPs, coloured as in **a**. 2-Aminoethlyphosphonate-specific cap residues are highlighted yellow, phosphite-specific cap residues are highlighted grey (PtxB) or purple (PhnD) and hypophosphite-specific cap residues are highlighted green. Conserved residues that form H-bonds with ligand oxygen atoms are highlighted in blue. The position in the alignment is shown above the sequence and residue numbering for each protein at row ends. For residue numbering, the red (or orange, see below) value corresponds to the residue numbering of the structures, as the predicted N-terminal signal peptides were not present in recombinant proteins, and the black number in parentheses is the residue number of the unprocessed proteins. The second residue (following the initiator Met1) in the recombinant proteins used in this study is shown in red and bold in the sequence; in the case of proteins not directly studied here, the equivalent residues/residue numbers are shown in orange. For full details of the proteins included in this figure see Supplementary Table 3

other proteins from this family with known structures are C-P lyase-linked PhnDs, namely the *E. coli* 2AEPn binding protein (PDB:3P7I) (Ec_PhnD)[14, 15] and its close homologue from *Pseudomonas aeruginosa* (PDB:3N5L). A structural comparison between Te_PtxB and Ec_PhnD (RMSD cα: 1.56 Å) shows that whilst the residues that interact with the ligand oxygen atoms are conserved, the capping residues that confer phosphite specificity in Te_PtxB (Y208 and D22) are replaced with D205 and T14 in Ec_PhnD (Fig. 2d, h). These changes result in Ec_PhnD having a larger binding pocket required for phosphonate binding, with D205 and the adjacent E177 also providing hydrogen bond acceptors to the amino group of the 2AEPn (Fig. 2d), contributing to the low nanomolar binding affinity for this ligand[14, 15, 35]. This D/T/E cap is conserved in many other group 1 members, including the *P. aeruginosa* PhnD, indicating high-affinity 2AEPn binding, however, there are also PhnD homologues in other bacteria with alternative residues at these positions, reflecting the diversity of R1 side chains of potential environmental phosphonate ligands[14].

**Prochlorococcus ecotypes with two phosphite transporters.** *Prochlorococcus* is abundant in low phosphate ocean regions but cannot grow on phosphonates[12, 35, 43], despite some ecotypes having homologues of PhnYZ, which have been shown to oxidise

2AEPn in vitro[44, 45]; therefore, phosphite could be an important P source to this globally significant oceanic bacteria. The gene encoding Pm_PhnD in *P. marinus* MIT9301 is not responsive to P-limitation and is not genomically linked to known phosphite or phosphonate utilisation genes[12, 43, 46], nevertheless, we and others[35] have shown that it binds phosphite with high affinity. Some *P. marinus* ecotypes also have a *ptxABCD* operon, encoding a PtxABC-transporter and a phosphite dehydrogenase[12], indicating that these bacteria have a second phosphite uptake system. Based on protein phylogeny (Fig. 5a and ref. [12]), these two types of phosphite binding protein not only branch separately from C-P lyase-linked PhnD and HtxB homologues, but also from each other, indicating that they represent phylogenetically separate phosphite transporters. Conversely to the phosphite-specific PtxB proteins discussed above, the *P. marinus* MIT9301 PtxB (Pm_PtxB) has previously been reported to have similar low micromolar binding affinities for phosphite, MPn and EPn, measured by ITC[35]. To try to explain this, we determined the structure of the Pm_PtxB/phosphite complex (Table 4), which shows that Pm_PtxB has the same overall fold as Te_PtxB, with complete conservation of the amino acids that coordinate the ligand, including the P-H...π interaction with Y206 (Supplementary Fig. 16). We calculated the volume of the binding pocket in Pm_PtxB and found that it is 23–29% larger than those of the

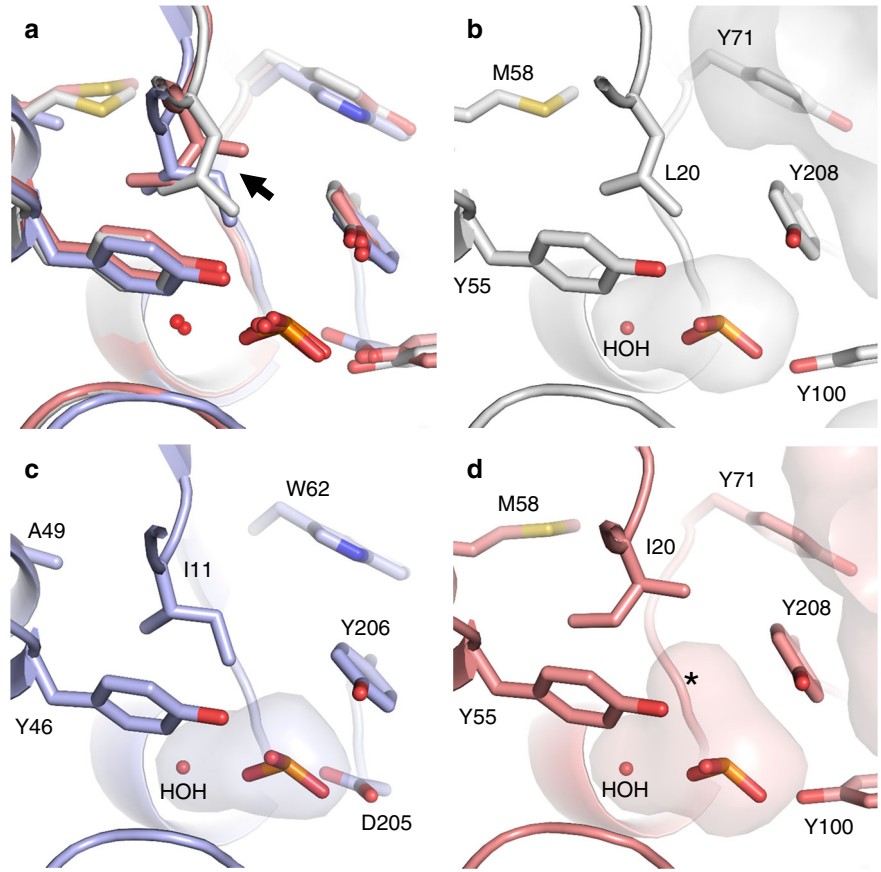

**Fig. 6** The volume of the binding pocket in Pm_PtxB is ~25% bigger than that of Te_PtxB and Pm_PhnD. Superposition of the Pm_PtxB/phosphite structure (pink) with those of Te_PtxB (grey) and Pm_PhnD (blue) (**a**) shows how a sequence difference (black arrow) in the binding pocket increases the volume of the cavity in Pm_PtxB. In Te_PtxB (**b**) and Pm_PhnD (**c**) Leu20 and Ile11 (respectively) provide a hydrophobic surface for the ligand to pack against within the binding pocket. In Pm_PtxB (**d**) Ile20 adopts an alternative rotamer making the hydrophobic region of the pocket slightly larger, as indicated by the asterisk (*****)

phosphite-specific PBPs (Supplementary Table 1). Comparison of the Pm_PtxB binding pocket with that of Te_PtxB (RMSD cα: 0.75 Å) and Pm_PhnD (RMSD cα: 1.52 Å) shows that this expansion is due to Ile20, which is conserved in Pm_PhnD (Ile11) and a leucine (Leu20) in Te_PtxB, as it adopts a different rotamer that forms a small hydrophobic cavity next to the capping tyrosine that is absent in the other proteins (Fig. 6). This seems to be due to several sequence changes in the second shell of amino acids within lobe 1 that create space for the Ile to rotate (Fig. 6). We were unable to measure the dissociation constant of Pm_PtxB for P-ligands here as the protein showed evidence of aggregation in the MST assay. Thus, from the structural similarities with Te_PtxB we can speculate that Pm_PtxB is most likely specific for phosphite, however, due to the extra space in the binding pocket could also bind simple phosphonates, in agreement with the previously reported ITC binding data[35]. The unknown substrate of the *Prochlorococcus* PhnYZ system, encoded downstream of the *ptxABCD* genes[12, 44], may also be a ligand of Pm_PtxB. Therefore, some *Prochlorococcus* ecotypes appear to have two independent phosphite transporter systems, and may also be able to take up some simple phosphonate species. The PhnDCE transporter is ubiquitous in *Prochlorococcus* and is also present in numerous marine *Synechococcus* strains, but its in vivo function is unknown as only ecotypes that have PtxABCD can grow on phosphite as a sole P source[12]. Further studies are therefore required to ascertain the biological role of phosphite uptake by the PhnDCE transporter and the in situ substrates of the PtxABC transporter in this important primary producer.

## Discussion

We have determined structures of PBPs from reduced phosphorus compound ABC transporter systems in complex with phosphite and hypophosphite. The structures reveal that ligand specificity, measured here by thermophoresis, arises from a combination of steric considerations and the interaction between a ligand P-H hydrogen atom and an aromatic π system in the binding pocket. In phosphite binding PBPs a strictly conserved tyrosine residue provides the π system, and in HtxB an invariant tryptophan makes the equivalent interaction with hypophosphite. We identified similar P-H…π interactions in 12 small-molecule crystal structures within the Cambridge Structural Database (Supplementary Table 2; Supplementary Fig. 17), however, such interactions with reduced phosphorus species could not be identified in protein structures in the PDB. It thus appears that, although C-H…π, N-H…π and cation-X…π bonds are well characterised in proteins[40, 41, 47–50], the structures reported here represent the first observations of P-H…π interactions in protein/ligand complexes.

Phosphite is a useful P source for biotechnological, industrial and agricultural applications. PtxD has been introduced to *Arabidopsis*, tobacco and rice, allowing the use of phosphite as a dual fertilisation and weed control system[28]. PtxD and phosphite are also used as selectable markers in bacteria[20, 34], yeast[24], plants[25] and algae[26]; to provide competitive advantage in industrial fermentation processes[27]; and in NADH-dependent biocatalysis for in situ cofactor regeneration[29]. So far these applications have only exploited PtxD, relying on non-specific phosphite uptake

that requires media phosphite concentrations of ≥0.1 mM. As phosphite is cheap it can be provided at high concentrations, but our binding data suggest that introducing a high-affinity transporter would reduce the phosphite requirement in these applications. Furthermore, some technologies require that phosphite uptake is highly specific. For example, a recent study by Hirota et al.[34] described the use of phosphite-dependent growth as a biocontainment strategy, which requires a phosphite transporter that does not transport phosphate. The authors introduced PtxD and the *Ralstonia* sp. 4506 PtxABC transporter into an *E. coli* strain devoid of its native phosphate acquisition systems, resulting in a strain that grew on phosphite, but also phosphate when the ligand was supplied at 1 mM. This concentration is several orders of magnitude higher than environmental phosphate concentrations, and based on the micromolar $K_d$ for phosphate binding to PtxB, determined in our study (Table 1), it is not surprising that phosphate supported growth. In the same study, addition of the *P. stutzeri* WM88 HtxBCDE hypophosphite transporter to the same *E. coli* strain did not allow growth on phosphate, as predicted by our binding studies which show that HtxB does not bind phosphate (Supplementary Fig. 5), but did allow growth on phosphite, which we would not have anticipated. Hirota et al.[34] show that the affinity of the HtxBCDE hypophosphite transporter is clearly high enough to support uptake in vivo when phosphite was supplied at 1 mM, but not when lowered to 5 μM, which still exceeds the highest known environmental phosphite concentrations. We did not detect significant phosphite binding by recombinant HtxB, which, from our structural evidence, is consistent with the architecture (Fig. 1f) and size (Supplementary Table 1) of the HtxB binding pocket. This suggests that for HtxB to bind phosphite the protein must undergo a conformational change or the mode of binding of the ligand would have to be different to that with hypophosphite.

In summary, we have determined the molecular basis for high affinity and specific phosphite and hypophosphite uptake in transporters from globally important and abundant terrestrial and marine microorganisms. The structures of two distinct types of phosphite-binding protein and a hypophosphite-binding protein reveal that a P-H…π interaction ensures high-affinity binding of specific ligands, representing a universal mechanism in these PBPs for recognising ligands with a P-H bond. The initial encounter with phosphite or hypophosphite in the bacterial periplasm is the first step in assimilating these potentially important P species, which are increasingly recognised as important constituents of the global phosphorus redox cycle[51] that may influence the biogeochemistry and productivity of the oligotrophic oceans and terrestrial ecosphere[1–12].

## Methods

**DNA manipulation.** Primers and plasmids used in this study are provided in Supplementary Tables 4 and 5 respectively. *Escherichia coli* JM109 competent cells (Promega, UK) were used for all standard cloning steps. The *Trichodesmium ptxB* gene (Tery_0366, Te_*ptxB*) minus predicted N-terminal signal peptide coding sequence (nucleotides 1–81 = amino acids 1–27, SignalP 4.1 Server[52]) and stop codon was amplified from *Trichodesmium erythraeum* IMS101 (Prufert-Bebout et al.[53], obtained from NCMA, USA) genomic DNA using Q5 DNA polymerase (New England Biolabs, UK) and primers Te_ptxB-F and Te_ptxB-R and cloned into the NdeI and XhoI sites of pET21a(+) (Novagen, UK). The *Prochlorococcus marinus* MIT 9301 *phnD* (P9301_07261, Pm_*phnD*) and *ptxB* (P9301_12511, Pm_*ptxB*) and *Pseudomonas stutzeri* WM88 (Ps) *ptxB* and *htxB* genes with their N-terminal signal peptide coding sequences[52] (Pm_*phnD* = nucleotides 1–72, amino acids 1–24; Pm_*ptxB* = nucleotides 1–63, amino acids 1–21; Ps_*ptxB* = nucleotides 1–69, amino acids 1–23; *htxB* = nucleotides 1–99, amino acids 1–33) and stop codons removed were synthesised with codons optimised for expression in *E. coli* (Integrated DNA Technologies, Inc., USA) and sub-cloned into pET21a(+) as above. The Quik-Change II Site-Directed Mutagenesis Kit (Aglient Technologies, USA) was used to generate genes encoding point mutants. All genes were sequence verified (GATC Biotech) and confirmed to be in-frame with the C-terminal His6-tag.

**Protein production and purification.** Plasmids were introduced into *E. coli* BL21 (DE3) (Invitrogen, UK) for protein production. Cultures were grown in LB broth with 100 μg ml⁻¹ ampicillin shaking (250 rpm) at 37 °C. At an optical density at 600 nm of ~0.6, isopropyl-β-D-thiogalactopyranoside was added to a final concentration of 0.4 mM to induce expression and the cultures were incubated for a further 16 h at 18–20 °C. Cells were harvested by centrifugation at 12,000 × g for 20 min at 4 °C and suspended in binding buffer (25 mM HEPES (4-(2-hydro-xyethyl)-1-piperazineethansulfonic acid) pH 7.5, 0.5 M NaCl, 5 mM imidazole, EDTA-free protease inhibitor (Roche, UK)). Cell suspensions were sonicated on ice (6 × 30 s bursts with 30 s intervals between) and insoluble debris removed by centrifugation at 45,000 × g for 20 min at 4 °C. Proteins were purified by immobilised Ni-affinity chromatography on a 5 ml Chelating Sepharose™ Fast Flow resin column (GE Healthcare, UK). Bound protein was washed with binding buffer containing 20 mM imidazole and then eluted into 25 mM HEPES pH 7.5, 400 mM imidazole, 0.1–0.2 M NaCl. Proteins were further purified on a 22 ml Superdex S200 Increase column (GE Healthcare) in 25 mM Tris/HCl, 0.2 M NaCl pH 7.4 and eluted as single peaks. Peak fractions were pooled and concentrated to 2.5 ml and protein was buffer exchanged into 25 mM Tris/HCl pH 7.4 with 0–0.2 M NaCl using a PD-10 desalting column (GE Healthcare). Proteins were used immediately or stored at 4 °C.

**Thermophoresis.** The binding affinity of proteins for P ligands was determined using Microscale Thermophoresis (MST) with a Monolith NT.115 instrument (NanoTemper Technologies, Germany). Protein (100 nM) was labelled with RED-tris-NTA dye (Nanotemper Technologies) according to the manufacturer's instructions in 50 mM HEPES pH 7.4, 250 mM NaCl, 0.05% Tween-20. A volume of 10 μl of 100 nM labelled protein was mixed with 10 μl of ligand in 50 mM HEPES pH 7.4, 250 mM NaCl, 0.05% Tween-20. 4 μl of protein–ligand mixture was loaded into Premium grade capillaries (Nanotemper Technologies) and thermophoresis was measured at 22 °C for 22 s with 20% LED power and 40% infrared laser power. Data from three independent measurements were combined and analysed using the MO.Affinity Analysis software version 2.1 (Nanotemper Technologies), fitted to a single binding site model (Eq. 1) where [*l*] is the concentration of ligand and data plotted using Igor Pro version 7 (Wavemetrics Inc., USA).

$$f(l) = \text{Unbound} + \frac{(\text{Bound} - \text{Unbound}) \times \left( [l] + [\text{protein}] + K_d - \sqrt{([l] + [\text{protein}] + K_d)^2 - 4 \times [l] \times [\text{protein}]} \right)}{2 \times [\text{protein}]}$$

(1)

Equation 1 shows the single binding site model used to determine dissociation constants.

**Isothermal titration calorimetry.** ITC experiments were performed on a Nano ITC (TA Instruments, USA) at 22 °C. Purified Te_PtxB (200 μM) was exchanged into a buffer containing 50 mM HEPES, 250 mM NaCl, 0.05% Tween-20, pH 7.4 using a PD-10 desalting column. The ligand was dissolved in the same buffer, and then diluted to 2 mM. Each titration series consisted of 32 injections of 1.49 μl ligand into a 160 μl protein sample. Control experiments were carried out by injecting ligand into the buffer, and the resulting heat of dilution was subtracted from the binding isotherm data. The first injection was ignored in the final analysis. The raw ITC data were processed using NanoAnalyse Data Analysis V 3.7.0 (TA Instruments) and fitted to a single-site binding model. Data were plotted in Igor Pro version 7.04 (Wavemetrics Inc.).

**Circular dichroism spectroscopy.** Spectra were recorded with a JASCO-810 spectrometer (JASCO, UK). Protein (0.1 mg ml⁻¹) was in 5 mM sodium phosphate buffer, pH 7.4 at 25 °C. Spectra were recorded in a cuvette with a 0.1-cm path length. Spectra were recorded continuously, from 250 to 190 nm (50 nm s⁻¹, 1 nm increments, 4 s response, 6 accumulations).

**Crystallisation and structure determination.** Te_PtxB (10 mg ml⁻¹) was co-crystallised with 5 mM sodium phosphite or methylphosphonic acid (MPn) (pH 7.0) by sitting-drop vapor diffusion (200 nl: 200 nl, 290 K) in 0.2 M NaCl, 0.1 M phosphate-citrate buffer pH 4.2 and 20% (w/v) PEG 8000 or 0.2 M magnesium chloride, 0.1 M sodium acetate pH 5 and 20% (w/v) PEG 6000, respectively. Pm_PhnD (7 mg ml⁻¹) was co-crystallised using the same method, but in 0.1 M SPG (succinic acid, sodium dihydrogen phosphate and glycine) buffer pH 8 and 25% (w/v) PEG 1500 or 0.1 M MMT (DL-malic acid, MES and Tris base) buffer pH 8 and 25% (w/v) PEG 1500, for the phosphite or MPn complexes, respectively. Apo Ps_PtxB (11 mg ml⁻¹) was crystallised using the same method, but in conditions containing 0.1 M tri-sodium citrate pH 5.5 and 20% (w/v) PEG 3000 and in the presence of 5 mM ethylphosphonate (EPn), no electron density for which was observed in the resulting structure. Crystals of Ps_PtxB (11 mg ml⁻¹) in complex with phosphite and MPn were grown at pH 5 and pH 4, respectively, from 0.1 M MMT buffer (DL-malic acid, MES and Tris base) and 25% (w/v) PEG 1500. HtxB (7 mg ml⁻¹) was co-crystallised with 3 mM sodium hypophosphite using the same method, but in 0.2 M sodium/potassium tartrate, 0.1 M bis-tris propane pH 8.5 and 20% (w/v) PEG 3350. For all the above proteins, thin, rod-shaped

crystals grew within 48 h (290 K). Conversely, long rod-shaped crystals of Pm_PtxB (5 mg ml$^{-1}$) in complex with phosphite (5 mM) took a number of weeks to grow at 290 K using the same method as the other proteins in conditions containing 0.1 M magnesium chloride, 0.1 M sodium citrate pH 5.0 and 15% PEG 4000. All crystals were harvested and cryoprotected in their mother-liquor with an additional 25% ethylene glycol or glycerol before plunge cooling in liquid nitrogen.

Crystals were mounted on a 100 K cryostream at the Diamond Light Source and for each structure data were collected from a single crystal at a single wavelength (for details see Tables 2 and 4). Data were autoprocessed using Xia2[54] and a resolution cut-off was applied using CC$_{half}$ (for processing statistics see Tables 2 and 4). The structure of the Te_PtxB/phosphite complex was determined using the automated molecular replacement pipeline, Mr. Bump[55]. The best solution was produced by Phaser[56] using a homology model generated by MolRep[57] from the structure of E. coli PhnD (PDB:3P7I)[14]. Automated model building with Buccaneer[58] produced an initial model comprising two copies of Te_PtxB in a back-to-back arrangement. Several rounds of manual model building and refinement were carried out in COOT[59] and Refmac5[60], before validating the final model with Molprobity[61]. The structures of the other complexes were determined by molecular replacement with Phaser[56] using a monomer of the highest-resolution structure of Te_PtxB or Pm_PhnD available at the time. The models were built, refined and validated using the same methods as those used with the Te_PtxB/phosphite complex, but the three structures that were sub-1.5 Å resolution (Te_PtxB/MPn, Pm_PhnD/phosphite and Ps_PtxB/MPn) were refined with anisotropic B-factors. The apo structure of Ps_PtxB was determined by molecular replacement using a model of the open structure of the protein, the conformation of which was based on the structure of the H157A mutant of E. coli PhnD. Phaser successfully found the six subunits that were contained within the asymmetric unit, which were subjected to ridged body refinement in PHENIX[62] to improve the fit to the electron density, prior to the same process of model building and refinement that was applied to the other structures (for refinement statistics see Tables 2 and 4).

In the structure of each ligand complex, clear electron density for the ligand was present in the binding pocket. However, as the three P ligands under study have similar chemical structures to each other and to phosphate, careful assignment of the ligand to the difference density was required. For each structure the relevant ligand was refined in the density in the orientation that made most chemical sense, considering van der Waals contacts and hydrogen bonds. These refinements were compared to controls, in which, for example, the phosphite was replaced with phosphate, or the methylphosphonate was refined with the methyl group in different orientations. In each case the controls confirmed that the ligand had been modelled correctly as evidenced by the B-factors, difference maps and relevant overall omit maps (Supplementary Figs. 12 and 13), which were calculated using SFCHECK (CCP4[63]). Ligand coordinate files and dictionaries are all generated in JLigand[64]. Superpositions were carried out in Superpose[65] by secondary structure matching of the Cα atoms. The volume of the ligand-binding pocket in each protein complex was analysed using the CASTp server using a probe radius of 1.4 Å[66] and UCSF Chimera[67]. Analysis of the crystal packing in each structure using the PISA sever[68] showed that no more than 5% of the total surface area of each protein was buried between neighbouring chains in the crystals, suggesting that all the proteins are monomeric, which agrees with the elution volume from size exclusion chromatography.

**Bioinformatics.** Homologues of PhnD/PtxB/HtxB proteins were identified by BLAST searches[69]. Sequence alignments were performed using ClustalW[70]. The phylogenetic tree was made with Geneious version 10.0.2 (http://www.geneious.com)[71]. For details of proteins used for these analyses see Supplementary Table 3.

**Data availability.** Atomic coordinates and structure factors for the reported crystal structures have been deposited in the RCSB Protein Data Bank (www.pdb.org) with accession codes: 5JVB (Te_PtxB/phosphite), 5LQ1 (Te_PtxB/MPn), 5LQ5 (Pm_PhnD/phosphite), 5LQ8 (Pm_PhnD/MPn), 5O2K (apo Ps_PtxB), 5O2J (Ps_PtxB/phosphite), 5O37 (Ps_PtxB/MPn), 5ME4 (HtxB/hypophosphite) and 5LV1 (Pm_PtxB/phosphite). Uniprot accession codes of proteins described in this study are given in Supplementary Table 3. The authors declare that the data supporting the findings of this study are available from the corresponding author upon reasonable request.

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

## Acknowledgements

This work was supported by grants BB/M000265/1 (to C.N.H.) and BB/M011305/1 (to T.S.B.) from the Biotechnology and Biological Sciences Research Council UK and grant 338895 (to C.N.H.) from the European Research Council. This work was also supported in part by the European Community's Seventh Framework Program (FP7/2007–2013) under Grant Agreement 283570 (BioStruct-X) and the Photosynthetic Antenna Research Center (PARC), an Energy Frontier Research Center funded by the U.S. Department of Energy, Office of Science, Office of Basic Energy Sciences (Award number DE-SC0001035). PARC provided partial support for C.N.H. and C.B. D.P. acknowledges a PhD studentship jointly funded by the Natural Environment Research Council and the A.G. Leventis Foundation. We thank the Diamond Light Source for beam time under BAG applications MX-8987 and MX-12788, and the beamline scientists on i03, i04, i04-1 and i24 for assistance with data collection. The authors also thank Professor G. Michael Blackburn and Dr Jack W. Chidgey (University of Sheffield), Dr Russell Viner (Syngenta), Professor C. Mark Moore (University of Southampton) and Dr Benjamin Van Mooy (Woods Hole Oceanographic Institution) for useful discussions relating to this work.

## Author contributions

A.H. and T.S.B. conceived the study. C.B., N.B.P.A., C.N.H. and A.H. designed the experiments. D.P. and A.H. generated expression plasmids. N.B.P.A., B.S., A.A.B. and A.H. purified proteins. N.B.P.A. acquired and interpreted binding data and did CD spectroscopy. C.B., B.S. and A.H. prepared crystals. C.B. collected and processed diffraction data, determined structures and performed structural analysis. B.S. assisted with binding data acquisition, diffraction data processing and structure determination for Ps_PtxB. P.J.B. performed structural analysis. A.H. performed bioinformatics. C.B. and A.H. wrote the paper. N.B.P.A., T.S.B., P.J.B. and C.N.H. edited and proofread the manuscript. All authors read and approved the final paper.

## Additional information

**Competing interests:** The authors declare no competing financial interests.

