## [Peer Review file · Nature Communications]

Reviewers' comments:

Reviewer #1 (Remarks to the Author):

Bisson and colleagues are presenting the structures of phosphite and hypophosphite periplasmic binding proteins (PBP) that are part of ABC importers that facilitate their uptake in bacteria. Their structures reveal a P-H... π interaction. Using binding data they show that the proteins are selective for phosphite and hypophosphite over phosphate.

The structures are not novel as many PBPs have been determined and published in the PDB. The authors comment on the existence of a novel P-H... π interaction based on their crystal structures. Such interactions have been observed in the structure of small molecules but not proteins. This is the first report and very interesting.

I have certain reservations especially for the Te_PtxB and Pm_PhxD complexes since they have been grown in a phosphate buffer. How can they authors be sure that the densities correspond to phosphite or methylphosphonate? The densities in Fig S4 could easily fit a phosphate group. In addition the authors do not show any Fo-Fc omit maps but only the final refined density. Have they also determined the apo structures since E. coli contains significant amount of phosphate? Different orientations could also be fit within the current densities.

In Fig S4 a, c and f extra density appears to be above the P atom that is not modelled. Anisotropic refinement for the data with resolution better than 1.5Å should have been used.

No mutagenesis studies have been performed to further investigate the aromatic residues involved in substrate binding and further validate the P-H... π interaction.

The fitting of the MST data in Fig S2a is not very convincing especially for the methylphosphonate and hyperphosphite. I am also surprised that the authors labeled and measured HtxB in PBS (for MST) since phosphate could be competing with the other substrates. I do not think that the data in Table 1 for HtxB are reliable or conclusive and the comment in line 183 "no binding of phosphite to Ps_HtxB could be detected" is not very valid; they should repeat the measurement in PBS free buffer.

The authors should also try sulphates since it has the same geometry as phosphates rather than formate.

PBPs are not enzymes as stated in line 173

Reviewer #2 (Remarks to the Author):

Bisson and co-workers provide a compelling story for molecular recognition of phosphite and hypophosphite by PhnD-like periplasmic binding proteins. These are essential gateway proteins for ABC-transporters that are used by bacteria to recognize and import these molecules as alternative sources of inorganic phosphate. The specificity of several of these proteins for phosphite and hypophosphite over organophosphonates is demonstrated by measured K_d values and X-ray crystal structures of several complexes. The latter reveal a novel P-H – π -electron interaction (in the form of Tyr or Trp residues) that appears to be used to distinguish phosphite and hypophosphite from organophosphonates. Such a mechanism does not appear to be known in the literature for protein-ligand interactions. Considering that oxidation of phosphite and hypophosphite to Pi by marine plankton represents a major contribution to the global phosphorus cycle, revealing this specificity mechanism is a significant contribution to the field. I believe in principle this work merits publication in Nature Communications. However, I have some concerns about the characterization of ligand binding to the PhnD-like proteins. I think this can be easily be addressed, but it will mean more experimental work. These and other comments are given below:

Comments for the main text

Line 54. This sentence should be more specific. The periplasmic binding protein PhnD was characterized. Phosphonate transporter is too vague.

Lines 86 to 88. Delete this sentence or explain more specifically how previous assays of P-ligand binding are not valid. Ironically, in my opinion, the authors should be more critical of the binding assay used in their studies (see below).

Line 104. Why did the authors choose to use microscale thermophoresis to determine binding constants when it wasn't necessary or does not appear to offer any significant advantages over ITC? Indeed, ITC would have provided additional interesting information, such as entropy and enthalpies of binding.

Additionally, can the authors demonstrate that fluorescent labelling of the PhnD-like proteins does significantly impact their functions? Periplasmic binding proteins undergo large conformational changes upon ligand binding. I would think that covalently labelling multiple Lys side chains with fluorophores might risk hindering this conformational change, or worse, prevent it altogether. At the very least this requires a comparison to binding constants obtained by ITC using unlabelled proteins (in which case, why not simply use ITC and avoid this uncertainty altogether).

Interestingly, Pm_PtxB was described as 'unstable' in the MST assay (line 246). What exactly 'unstable' means in this context is not clear, but I expect chemical labelling might be an issue. Why not try ITC with Pm_PtxB?

Lines 158-159. I think a semicolon is needed in this sentence as follows "...capping tyrosine (Y208/Y229); despite this movement...."

Line 332. I think reference 2 is a bit out of date as a general review of Pn catabolism. I think a new review came out recently in Chem. Rev.

Lines 159-161 - This is a strange sentence that needs rewriting or expansion. If the P-CH₃ - Tyr interaction is destabilizing, why is a close interaction (2.1 Å) maintained with the methyl group and the face of the Tyr side chain? This appears to be even closer than the P-H - Tyr interaction observed in the phosphite complex (2.6 Å, line 140)!

General comment about the P-H - Tyr / Trp interactions. I think mutagenesis should be used to probe this interaction. The obvious changes are Tyr to Phe and Tyr to Ala in the case of the phosphite binding proteins, and Trp to Tyr, Phe, and / or Ala in the case of hypophosphite binders. Can we learn something from the electron richness of the pi-system? Can ligand specificity be changed? This is the main point of the paper afterall!

Figure 3 (line 657), frames d, e, and g. Why are the His side chains shown in ionized form? Would not the conjugate acids, with a formal positive charge, be better suited to form an interaction with a negatively charged phosphoryl oxygen?

Comments for the SI

Supplementary Figure S2: 'hyperphosphite' needs to be corrected.

Reviewer #3 (Remarks to the Author):

This manuscript presents a series of high-resolution crystal structures of bacterial periplasmic proteins that function as phosphite or hypophosphite binding proteins. The crystallized proteins are part of tripartite complex transporters, also including a bona fide permease and an ATPase, and which are present in specific bacterial species living in P limiting environments. The binding affinity (K_d values) of the proteins crystallized for a series of putative ligands was determined, using the relative polypeptides purified in *E. coli*, by microscale thermophoresis. The authors provide evidence that the purified polypeptides are high-affinity and rather specific phosphite or hypophosphite binding proteins. They further provide structural evidence strongly suggesting that specificity is achieved by a combination of steric selection and the presence of a novel P-H... π interaction between the ligand and an aromatic residue that caps the ligand-binding pocket. They finally discuss their results from an evolutionary perspective and in relation to plausible uses of phosphite or hypophosphite binding proteins in biotechnology.

Major comment

The work presented is very good, rigorous, to the point, and leads to some clear cut conclusions. It is also novel and original as no similar proteins have been analyzed structurally. The part of results related to the evolution of these proteins is also quite satisfactory. So overall this is a very decent research work that is worthy of publication in a very good journal. My problem is whether it is proper for publishing it in *Nature Communications*, as it is, given that all conclusions drawn come from a purely structural study using heterologously expressed polypeptides. The work lacks *in vivo* functional evidence for the role of the proteins. What surprises me is that there was no attempt to mutate some of the critical residues determined and see what happens. The authors have put the structural basis on how specificity in these proteins is determined via their different substrate affinities for specific ligands. It would be quite simple to construct relevant mutant versions of these proteins, and even try to interchange specificities through rational mutational design, in order to support their conclusions. In addition, I would have liked to see a discussion, or even additional experiments, on whether the *in vivo* transport specificity of the relevant ABC transporters is solely determined by the specificity of ligand binding of the peripheral proteins crystallized herein, or whether the permease unit of these transporters contributes to specificity. Finally, in the phylogenetic analysis I would have liked to see a deeper analysis on including some out-groups and a discussion what happens in microbial or plant systems that do not possess homologous proteins with the ones described herein.

Conclusion

A mutational analysis of the relevant transporters, accompanied with relative binding assays, as well as, a discussion on the role of the permease unit of the relevant ABC transporters, are minimal essential requirements before publication in *Nature Communications*.

Point by point response to Reviewer's comments:

Reviewer #1 (Remarks to the Author)

Bisson and colleagues are presenting the structures of phosphite and hypophosphite periplasmic binding proteins (PBP) that are part of ABC importers that facilitate their uptake in bacteria. Their structures reveal a P-H... π interaction. Using binding data they show that the proteins are selective for phosphite and hypophosphite over phosphate.

The structures are not novel as many PBPs have been determined and published in the PDB. While the Reviewer is right that many PBP structures have been determined previously, the substrate specificity of ABC-transporters is conferred by the PBP, and so our structures are novel in terms of the ligand as no previous structures of phosphite or hypophosphite binding proteins have been deposited.

The authors comment on the existence of a novel P-H... π interaction based on their crystal structures. Such interactions have been observed in the structure of small molecules but not proteins. This is the first report and very interesting.

I have certain reservations especially for the Te_PtxB and Pm_PhxD complexes since they have been grown in a phosphate buffer. How can they authors be sure that the densities correspond to phosphite or methylphosphonate? The densities in Fig S4 could easily fit a phosphate group.

Great care was taken to interpret the ligand density correctly, especially, as the Reviewer rightly points out, some of the crystallisation conditions contained phosphate buffers. In order to confirm our interpretation, we carried out a ligand validation exercise, which was not described in the original manuscript. We did this as follows:

For the Pm_PhxD (1.4 Å resolution; crystallised in phosphate containing buffer) and Pm_PtxB (2.1 Å resolution) complexes with phosphite, we modelled and refined a phosphate moiety into the ligand density. This led to a -3.0 σ difference peak and a high B-factor at the fourth oxygen position, as well as distorted O-P-O bond angles and severe clashes between the O4 position and the capping tyrosine (~2.7 Å). Restraining the bond lengths and angles of the phosphate during refinement also produced positive difference peaks on two of the other phosphate-oxygen atoms as the anti-bumping restraints in the refinement program push the ligand away from the capping tyrosine, causing them to be positioned off-centre in the electron density.

For the complexes that contained MPn, where the referee suggests that different orientations could also fit within the density, we refined the ligand in the Pm_PhxD/MPn complex (1.45 Å resolution) in several different orientations. The B-factors and difference density peaks in the resulting maps were consistent with the methyl group pointing towards the capping tyrosine, which also agrees with the tight network of hydrogen bonding around the three MPn oxygen atoms.

We have added a new figure to the SI materials (Figure S9) that shows the maps generated from these validations and a table of B-factors for the trial ligands after refinement. We have also added a paragraph to the materials and methods describing the approach we used. We have done these validations in both high and medium resolution structures and obtained the same result. We hope this serves to convince the Reviewer that our ligand interpretation is correct.

In addition, the authors do not show any Fo-Fc omit maps but only the final refined density.

Apologies, this was an oversight. We have now used sfcheck (ccp4) to generate overall omit maps for each structure and have amended Figure S8 to show the section of map around the ligand.

Have they also determined the apo structures since *E. coli* contains significant amount of phosphate? Different orientations could also be fit within the current densities.

The Reviewer rightly wonders whether phosphate might have co-purified with the protein from the expression host. However, we had numerous attempts to crystallise the apo form of these proteins during this study and in each case no crystals could be formed. This suggested to us that phosphate, either from the crystallisation buffers or the expression host, does not bind to the proteins with high affinity, as otherwise a closed complex would be present, which in our experience crystallizes readily. This is further supported by the binding data. Since the manuscript was reviewed we have managed to obtain a crystal structure of apo-PtxB from *Pseudomonas stutzeri*, which was expressed and purified in the same way as the other proteins in this study. This suggests that phosphate does not co-purify from the expression host. We have added this structure and a comparison with the closed ligand-bound structures of the protein to the paper in a new section (see results from line 289-321).

In Fig S4 a, c and f extra density appears to be above the P atom that is not modelled.

Please see our comments above concerning the ligand validation carried out on these structures and refer to Figure S9.

Anisotropic refinement for the data with resolution better than 1.5Å should have been used.

The two structures that were described in the manuscript with a resolution of better than 1.5Å were refined with anisotropic B factors, but this was mistakenly omitted from the methods. This has now been amended. The new structure of the Ps_PtxB /MPn complex has also been refined with anisotropic B factors.

No mutagenesis studies have been performed to further investigate the aromatic residues involved in substrate binding and further validate the P-H... π interaction.

To explore the importance of the P-H... π interaction for phosphite binding we have now produced mutations in Te_PtxB (Y208A and Y208F). These mutations did not disrupt the secondary structure of the protein (as shown by circular dichroism, Figure S6), and when studied by thermophoresis showed a 3 orders of magnitude difference in K_d between the 'wildtype' and mutant protein (Figure S7 and Table 3). In addition, we produced mutants of HtxB (W52A, W52F and W52Y), these mutations obliterated the affinity for hypophosphite as shown in Figure S7 and Table 3. We have added new paragraphs to the relevant sections to put these findings into a structural context (see lines 178-197 and 228-237 for Te_PtxB and lines 273-283 for HtxB).

The fitting of the MST data in Fig S2a is not very convincing especially for the methylphosphonate and hypophosphite. I am also surprised that the authors labeled and measured HtxB in PBS (for MST) since phosphate could be competing with the other substrates. I do not think that the data in Table 1 for HtxB are reliable or conclusive and the comment in line 183 "no binding of phosphite to Ps_HtxB could be detected" is not very valid; they should repeat the measurement in PBS free buffer.

The Reviewer's comments were noted and we understand their concern regarding the phosphate in the assay buffer. We have now labelled the proteins and repeated all the experiments in phosphate-free buffer (50 mM HEPES, 250 mM NaCl, 0.05% Tween-20 pH 7.4) to ensure consistency. We have also now carried out all the MST experiments using a His₆ labelling technique with TRIS-NTA-RED dye. This labelling technique does not covalently modify the protein, as this dye labels only at the His₆ of the recombinant protein, which, as we can see from the crystal structure, lies remotely from the binding pocket. This limits issues arising from the label interfering with conformational changes upon ligand binding. Pleasingly, with the change of buffer

and labelling technique, the new binding curves (Figure S2) show lower error bars and provide high quality data for fitting to obtain the K_d for binding. In addition, the K_d for all proteins and ligands were very similar to the values reported in the prior version of this manuscript (Table 1).

With respect to HtxB being unable to bind phosphate, we altered the buffer as suggested and performed repeated titrations with increasing concentrations of phosphate and saw no evidence for binding up to 10 mM phosphate. The raw data for this 'binding' check is summarized in Figure S4. It shows that while there is a significant change in the level of thermophoresis when HtxB binds its native substrate hypophosphite, there is little to no change in the presence of high concentrations of phosphate, well below the level of signal change to represent binding as demonstrated by the instrument manufactures. This is in agreement with the recent study of Hirota *et al.* (*Sci. Rep.* 7, 44748 (2017)), who found phosphate is not taken up by HtxBCDE even when provided at the not physiologically relevant concentration of 1 mM.

The authors should also try sulphates since it has the same geometry as phosphates rather than formate. The reason we tested formate binding is that it has previously been suggested that HtxA may have a promiscuous substrate-specificity as it had residual activity with formate, but we agree with the Reviewer that its geometry is different to the P ligands studied here so we have removed this from the manuscript.

We have also considered trying ligands with similar geometries to phosphite such as sulphite/arsenite to see if we can ascertain how the proteins discriminates between these similar anions. However, this would be a considerable amount of additional work and would require very high resolution crystal structures alongside appropriate ligand-binding data to gain a clear understanding (e.g. for phosphate/arsenate discrimination as reported by Elias *et al.* *Nature.* 491, 134-137 (2012)). We feel such a significant undertaking would constitute a separate study and is too much to include in the current paper.

PBPs are not enzymes as stated in line 173.
This has been amended.

Reviewer #2 (Remarks to the Author)

Bisson and co-workers provide a compelling story for molecular recognition of phosphite and hypophosphite by PhnD-like periplasmic binding proteins. These are essential gateway proteins for ABC-transporters that are used by bacteria to recognize and import these molecules as alternative sources of inorganic phosphate. The specificity of several of these proteins for phosphite and hypophosphite over organophosphonates is demonstrated by measured K_d values and X-ray crystal structures of several complexes. The latter reveal a novel P-H – pi-electron interaction (in the form of Tyr or Trp residues) that appears to be used to distinguish phosphite and hypophosphite from organophosphonates. Such a mechanism does not appear to be known in the literature for protein-ligand interactions. Considering that oxidation of phosphite and hypophosphate to Pi by marine plankton represents a major contribution to the global phosphorus cycle, revealing this specificity mechanism is a significant contribution to the field. I believe in principle this work merits publication in Nature Communications. However, I have some concerns about the characterization of ligand binding to the PhnD-like proteins. I think this can be easily be addressed, but it will mean more experimental work. These and other comments are given below:

Comments for the main text

Line 54. This sentence should be more specific. The periplasmic binding protein PhnD was characterized.

Phosphonate transporter is too vague.

We agree with the referee and this sentence has been amended to clarify that it was the PBP, specifically, that was characterized.

Lines 86 to 88. Delete this sentence or explain more specifically how previous assays of P-ligand binding are not valid. Ironically, in my opinion, the authors should be more critical of the binding assay used in their studies (see below).

The Reviewer makes a good point, we were less than generous in our initial manuscript when accounting for the apparent discrepancy between our binding data with phosphite-specific PBPs, the structure of Pm_PtxB with phosphite and the binding data reported by ITC in the previous study of Feingersch *et al.* (ISME J. 6, 827-834 (2012)). Since the first submission we have carried out further analysis of the Pm_PtxB structure and it has become apparent that the binding pocket in this PBP is on average ~26 % larger than the Te_PtxB and Pm_PhxD. This expansion is due to Ile20, which is conserved in Pm_PhxD and a leucine (Leu20) in Te_PtxB, as it adopts a different rotamer that forms a small hydrophobic cavity next to the capping tyrosine that is absent in the other proteins. This seems to be due to several sequence changes in the second shell of amino acids within lobe 1 that create space for the Ile to rotate. The extra space in the binding pocket could perhaps enable this PBP to bind MPn or EPn as well as phosphite, in agreement with previously reported ITC binding data. We have therefore deleted our previous statement and discussed this new analysis between lines 373 and 395 in the new version of the manuscript. We thank the Reviewer for bringing this oversight to our attention.

Line 104. Why did the authors choose to use microscale thermophoresis to determine binding constants when it wasn't necessary or does not appear to offer any significant advantages over ITC? Indeed, ITC would have provided additional interesting information, such as entropy and enthalpies of binding.

Thermophoresis (MST), although a relatively recently developed technique, has been validated repeatedly in the literature as a *bona fide* binding technique, with over 1250 publications using the technique since 2010. It rapidly and reproducibly produces binding data, using protein at multiple orders of magnitude lower concentration than required for ITC (nM as opposed to μ M-mM). Working at the nM concentration range is likely to reduce concentration artefacts such as self-assembly, dimerization and precipitation (which we discovered when we performed ITC with Te_PtxB). ITC requires mechanical stirring which can lead to mechanical stress on proteins, leading to denaturing and precipitation. Like ITC, MST is a solution based technique so is not liable to orientation based effects associated with surface based techniques such as surface plasmon resonance.

Also like ICT, MST can produce information on the thermodynamics of ligand binding. MST does this by recording the change in K_d over a temperature range and using a Van 't Hoff plot to calculate enthalpy. However, explanations of enthalpy and entropy of ligand binding is often a 'retroactive' activity. A useful perspective on the dangers of relying on thermodynamic data for insight on binding is highlighted in this article: Geschwindner *et al. J. Med. Chem.* 58, 6321–6335 (2015).

Additionally, can the authors demonstrate that fluorescent labelling of the PhnD-like proteins does significantly impact their functions? Periplasmic binding proteins undergo large conformational changes upon ligand binding. I would think that covalently labelling multiple Lys side chains with fluorophores might risk hindering this conformational change, or worse, prevent it altogether. At the very least this requires a comparison to binding constants obtained by ITC using unlabelled proteins (in which case, why not simply use ITC and avoid this uncertainty altogether).

This is a valid concern and to address it we have now carried out all the MST experiments using a His₆ labelling technique with TRIS-NTA-RED dye, as described above. This labelling technique does not covalently

modify the protein, as this dye labels only at the His₆ of the recombinant protein, which, as we can see from the crystal structure, lies remotely from the binding pocket. This limits issues arising from the label interfering with conformational changes upon ligand binding. This technique produced effectively the same results as the previous surface-lysine labelling technique.

To assuage the Reviewers concerns with our assay strategy, we have now performed ITC on Te_PtxB. This was technically challenging and required substantial optimization as it was found that at the high concentrations of protein required to observe a change in the heating rate on ligand binding to the protein there was a concurrent risk of protein precipitation. This precipitation does not occur without the mechanical mixing present in the ITC experiment. After optimizing detergent concentrations (to be the same as MST) and mixing rate, we determined the K_d for both phosphate and phosphite binding by ITC to be in the same order of magnitude as determined by our MST method. Along with the similarity of our Pm_PhnD binding constants to those determined by ICT in a previous study (Feingersch *et al.* *ISME J.* 6, 827-834 (2012)), we hope the Reviewer agrees the MST data is validated.

Interestingly, Pm_PtxB was described as 'unstable' in the MST assay (line 246). What exactly 'unstable' means in this context is not clear, but I expect chemical labelling might be an issue. Why not try ITC with Pm_PtxB?

The raw data for the MST showed unsmooth traces, indicative of protein precipitation or aggregation. Optimization with buffer and detergent changes did not alter these results. It is likely that the protein would also be precipitated or aggregated in an ITC experiment, especially at the high concentrations required for ITC and with mechanical stirring. Please see the above response for discussion of Pm_PtxB.

Lines 158-159. I think a semicolon is needed in this sentence as follows "...capping tyrosine (Y208/Y229); despite this movement...."

This whole paragraph has been reworked to explain more clearly how the position of the methyl group of the MPn is accommodated by an expansion of the binding pocket due to a movement of the capping tyrosine. Please also note that the Pm_PhnD structures have been renumbered to take into account the N-terminal signal sequence and to make the numbering consistent with the other structures. This renumbering changes Y229 to Y206. All numbers for this structure have been changed appropriately throughout the text, and also in the figures and the PDB deposition.

Line 332. I think reference 2 is a bit out of date as a general review of Pn catabolism. I think a new review came out recently in Chem. Rev.

The suggested review (Horsman & Zechel. Phosphonate Biochemistry. *Chem. Rev.* 117, 5704-5783, 2017) has now been cited (ref 10).

Lines 159-161 - This is a strange sentence that needs rewriting or expansion. If the P-CH₃ – Tyr interaction is destabilizing, why is a close interaction (2.1 Å) maintained with the methyl group and the face of the Tyr side chain? This appears to be even closer than the P-H – Tyr interaction observed in the phosphite complex (2.6 Å, line 140)!

We agree that this section of the manuscript was confusing. Further analysis of the binding sites of the proteins has shown that the ligand binding pocket increases in size by ~16% in the MPn complex, compared to the phosphite complex, with most of this expansion coming from the movement of the capping tyrosine residue. In the MPn complexes, this results in the methyl group lying close enough to the π system of the ring to form a C-H... π hydrogen bond. However, the geometry of the methyl to Tyr ring is such that the interaction would be sub-optimal, in comparison to the P-H... π interaction in the phosphite complexes, which appears to be ideally aligned for maximum interaction. This less than optimal arrangement for MPn is

seen in the structures of all three MPn complexes, which have all been determined to high resolution and we are confident of the position of the MPn ligand in each case. In addition, the mutation of the capping tyrosine to phenylalanine abolishes MPn binding suggesting that the network of intramolecular hydrogen bonds between the tyrosine and surrounding residues might be also a crucial component of the ligand binding pocket. We have altered the text to explain this in greater detail (lines 207-237) and added extra references to work on C-H... π interactions.

General comment about the P-H - Tyr / Trp interactions. I think mutagenesis should be used to probe this interaction. The obvious changes are Tyr to Phe and Tyr to Ala in the case of the phosphite binding proteins, and Trp to Tyr, Phe, and / or Ala in the case of hypophosphite binders. Can we learn something from the electron richness of the pi-system? Can ligand specificity be changed? This is the main point of the paper after all!

As stated above, we have now characterized all the capping residue mutants suggested by this Reviewer (Y to F and A in Te_PtxB; and W to A, Y and F in HtxB) and explained our observations in the context of the structures.

Both Reviewer 2 and 3 mention changing ligand specificity and rational mutational design. We believe this type of protein engineering/synthetic biology approach would be fascinating but would likely require multiple amino acid substitutions followed by corresponding ligand-binding and structural analysis, and as such would be a future study following on from this current work.

Figure 3 (line 657), frames d, e, and g. Why are the His side chains shown in ionized form? Would not the conjugate acids, with a formal positive charge, be better suited to form an interaction with a negatively charged phosphoryl oxygen?

It is true that a formal positive charge on the histidine in the binding pocket would be an ideal counter-ion for phosphite and we did consider this as a possibility. However, in the Te_PtxB and Pm_PtxB structures the histidine is positioned such that the imidazole nitrogen that does not interact with the ligand points directly at a main chain amino group (N-N distance ~ 3.2 Å). If the histidine carried two protons, the His-H to mainchain N-H distance would be ~ 1.5 Å, which is far too close, suggesting that the His is neutral and accepts a hydrogen bond from the main chain N-H.

In Pm_PhnD the situation is different as the histidine sidechain is rotated by $\sim 45^\circ$ due to an adjacent Ile to Val sequence difference. The imidazole nitrogen atom now points at a buried water molecule that is absent in the PtxB protein structures. In Pm_PhnD the histidine could therefore be positively charged, but we cannot confirm this from the network of hydrogen bonds around the water molecule, so this would be pure speculation. Also, if you consider the pH of the crystallisation of Pm_PhnD, which was pH 8, and notwithstanding an unusual pKa in the binding pocket, which we can't rule out, this is well above the pKa at which you would expect the histidine to be predominantly positively charged.

Finally, if you consider the environment in which these proteins function, sea water is slightly alkaline (\sim pH 7.5-8.4) and as the pH of the periplasm would be at a similar pH to the external environment this also suggests that the histidine would be neutral. Overall, there is no obvious source of a formal positive charge in the binding pocket to balance the phosphite, but perhaps you don't need one and the stabilisation energy of the closed form of the PBP, with its tight van der Waals packing and complex network of hydrogen bonds is enough.

Considering the above arguments, we have not amended the Figures, but we have added a paragraph to discuss these points at lines 161-168.

Comments for the SI

Supplementary Figure S2: 'hyperphosphite' needs to be corrected.

We have amended this typographical error here and throughout the text and thank the Reviewer for spotting it!

Reviewer #3 (Remarks to the Author)

This manuscript presents a series of high-resolution crystal structures of bacterial periplasmic proteins that function as phosphite or hypophosphite binding proteins. The crystallized proteins are part of tripartite complex transporters, also including a bona fide permease and an ATPase, and which are present in specific bacterial species living in P limiting environments. The binding affinity (K_d values) of the proteins crystallized for a series of putative ligands was determined, using the relative polypeptides purified in *E. coli*, by microscale thermophoresis. The authors provide evidence that the purified polypeptides are high-affinity and rather specific phosphite or hypophosphite binding proteins. They further provide structural evidence strongly suggesting that specificity is achieved by a combination of steric selection and the presence of a novel P-H... π interaction between the ligand and an aromatic residue that caps the ligand-binding pocket. They finally discuss their results from an evolutionary perspective and in relation to plausible uses phosphite or hypophosphite binding proteins in biotechnology.

Major comment

The work presented is very good, rigorous, to the point, and leads to some clear cut conclusions. It is also novel and original as no similar proteins have been analyzed structurally. The part of results related to the evolution of these proteins is also quite satisfactory. So overall this is a very decent research work that is worthy of publication in a very good journal. My problem is whether it is proper for publishing it in Nature Communications, as it is, given that all conclusions drawn come from a purely structural study using heterologously expressed polypeptides.

The work lacks in vivo functional evidence for the role of the proteins.

The purpose of our study was to understand the precise molecular basis of phosphite and hypophosphite recognition by the PBPs of these transporters and determine how the substrate specificity we have shown via MST for each is imparted by the architecture of the binding pocket. Functional evidence for the role of the proteins has been provided in a number of previous works, as described in the introduction to our manuscript. The majority of proteins analysed in our study are from globally important and abundant oceanic *Trichodesmium* and *Prochlorococcus* species. Genetic manipulation is not currently possible in either of these organisms so we could not make deletion strains and look for phenotypes, however the function of PtxABCD has been reported previously by growth on phosphite as a sole source of P and heterologous expression in another host (Polyviou *et al. Environ. Microbiol. Rep.* 7, 824-830 (2015); Martinez *et al. Environ. Microbiol.* 14, 1363-1377 (2012)). Similarly, in *Pseudomonas stutzeri* molecular genetic and biochemical analysis has indicated the roles of these transporters in phosphite/hypophosphite uptake (e.g. Metcalf and Wolfe. *J. Bacteriol.* 180, 5547-5558 (1998)). In addition, a recent study (Hirota *et al. Sci. Rep.* 7, 44748 (2017)) did express the full HtxBCDE transporter in an engineered strain of *E. coli* in which 8 separate mutations meant no other phosphate/phosphonate uptake systems or phosphite oxidising enzymes were present. To generate such a background strain from scratch would be very time consuming and we felt this was beyond the scope of our *in vitro* study, however this independent *in vivo* study complements our work in some

respects and a detailed discussion section is included in the revised manuscript (lines 423-441).

What surprises me is that there was no attempt to mutate some of the critical residues determined and see what happens. The authors have put the structural basis on how specificity in these proteins is determined via their different substrate affinities for specific ligands. It would be quite simple to construct relevant mutant versions of these proteins, and even try to interchange specificities through rational mutational design, in order to support their conclusions.

As stated above in the response to Reviewers 1 and 2, we have now characterized several mutants in the capping residue and explained our observations in the context of the structures. Please also see the response to Reviewer 2 for an explanation of why we have not undertaken further work to try and interchange substrate specificities.

In addition, I would have liked to see a discussion, or even additional experiments, on whether the *in vivo* transport specificity of the relevant ABC transporters is solely determined by the specificity of ligand binding of the peripheral proteins crystallized herein, or whether the permease unit of these transporters contributes to specificity.

The substrate specificity of bacterial ABC-importers is typically determined by the high affinity PBP component of the transporter, which delivers the ligand to the transmembrane permease that facilitates transport. There are also reports that orphan PBPs can bind their substrates and dock with single promiscuous permease and ATPase core transporter (e.g. Thomas. *Mol. Microbiol.* 75, 6-9 (2009)).

As mentioned above, the complete Ptx and Htx transporters have previously been expressed in heterologous hosts and shown to allow phosphite or hypophosphite transport/utilisation (e.g. Polyviou *et al. Environ. Microbiol. Rep.* 7, 824-830 (2015) and Hirota *et al. Sci. Rep.* 7, 44748 (2017)). If the Reviewer is suggesting we do an *in vivo* 'mix and match' experiment with different combinations of PBPs and permeases, while this may be informative we feel it is well beyond the scope of the current work, and, as mentioned above, it will ultimately be the PBP that governs substrate specificity.

Finally, in the phylogenetic analysis I would have liked to see a deeper analysis on including some out-groups. Our phylogenetic analysis was performed to show the relative relationship of the sub-groups to one another rather than to examine the evolutionary history, which is not straightforward in such a large family of proteins as the ABC-transporters, hence we did not generate a rooted tree using an out-group. As we are analysing many different PBPs from numerous bacterial species, choosing an out-group that would yield an informative tree is not straightforward, thus we have not altered the original tree.

and a discussion what happens in microbial or plant systems that do not possess homologous proteins with the ones described herein.

We do not understand the relevance of the question about what happens in microbial or plant systems that do not contain phosphite or hypophosphite utilisation systems. As discussed in the manuscript (and articles cited therein), in the absence of these systems organisms cannot grow using these reduced P compounds as the sole source of P. We have not added a specific discussion of this to the manuscript, but we have tried to make this point clearer in the text.

Conclusion

A mutational analysis of the relevant transporters, accompanied with relative binding assays, as well as, a discussion on the role of the permease unit of the relevant ABC transporters, are minimal essential requirements before publication in Nature Communications.

Please see above replies.

REVIEWERS' COMMENTS:

Reviewer #1 (Remarks to the Author):

The authors have addressed all my queries. They have included additional functional data and much improved MST measurements including mutagenesis. Since the first review, they have determined the structure of an apo SBD. They have also presented convincing validation electron density maps for the substrates.

Only two minor comments:

1. I think the apo structure should be presented at the beginning of the results section than later. It will flow much better.

2. The authors have missed the opportunity to do some more in depth analysis after the anisotropic refinement of the Te_PtxB/MPn, Pm_Phnd/phosphite and Ps_PtxB/MPn structures. It would have been interesting to include a supplementary figure with plotting the ellipsoids for the ligands and discuss a bit further the selectivity.

Reviewer #2 (Remarks to the Author):

The revised document by Bisson et al addresses the majority of my initial concerns, particularly with the MST binding assay and the mutagenesis experiments. I believe the paper is suitable for publication. The authors may want to consider the following comments.

Line 91-92: "...because their utilization requires fewer protein components ...compared to phosphonate breakdown...."

I'm not sure if this is a valid comparison of phosphite and phosphonates as Pi sources.

CP-lyase is not the only system that is used by bacteria to catabolize organophosphonates; there are also the CP hydrolases (eg: phosphonatease, phosphonoacetate hydrolase, etc) and the oxidative PhnY/PhnZ pathway. These are simple systems that do not require ATP and liberate Pi from phosphonates in 1-2 chemical steps; however, they are much more substrate specific relative to CP-lyase.

Line 100: "ligand specificity" mentioned as not being well studied.

Lines 157-158: "...suggests that...monoanionic phosphite is the predominantly bound form"

This builds off my earlier comment about the ionization state of the active site His. Could the ionization state be probed more directly using ³¹P-NMR spectroscopy of phosphite bound to PhnD? The one-bond J(PH) coupling constant is known to be different for dianionic (PHO₃²⁻) and monoanionic (PHO₃H⁻) forms:

Eykyn, T. R.; Kuchel, P. W. Scalar Couplings as pH Probes in Compartmentalized Biological Systems: ³¹P NMR of Phosphite. *Magn. Reson. Med.* 2003, 50 (4), 693–696.

And ³¹P-NMR has long been used to examine Pi and phosphonates bound to proteins:

Chlebowski, J. F.; Armitage, I. M.; Tusa, P. P.; Coleman, J. E. ³¹P NMR of Phosphate and Phosphonate Complexes of Metalloalkaline Phosphatases. *J. Biol. Chem.* 1976, 251 (4), 1207–1216.

I think this could be a relatively painless experiment (famous last words!) to apply to the best behaved (ie: most soluble, least prone to precipitation) PhnD variant. I will not insist upon this experiment for the paper, but it's worth considering. It might also be a way to probe the P-H aromatic pi system interaction.

Lines 182-183 and Table 3

The mutagenesis results are intriguing and not what I would have predicted!

The Y208A substitution in Te_PtxD was less detrimental than the more conservative Y208F substitution for phosphite binding. If deleting the hydroxyl of tyrosine prevents closing of the cap, the alanine substitution would do this as well, plus lose any pi interaction with the phosphite P-H. Any thoughts?

"1 to 2 orders of magnitude weaker" The Kd value changes from 0.17 uM in the WT to 135 uM in Y208A (800 fold) and 295 uM in Y208F (1735 fold). This is more than a 10 or 100 fold change.

Should also fix up the significant figures of the Kd values in Table 3 (eg: 51.3+/-1.76 should be 51+/-2).

Table 3 abbreviations (and elsewhere): I haven't seen Pst and HPst used as acronyms for phosphite and hypophosphite, respectively. Pst has been used previously in the literature, most prominently by Wanner, for the bacterial Phosphate Specific Transport system.

Hsieh, Y.-J.; Wanner, B. L. Global Regulation by the Seven-Component Pi Signaling System. *Curr Opin Microbiol* 2010, 13 (2), 198–203.

However, Pt and HPT have been used previously for phosphite and hypophosphite. See:

White, A. K.; Metcalf, W. W. The Htx and Ptx Operons of *Pseudomonas Stutzeri* WM88 Are New Members of the Pho Regulon. *J. Bacteriol.* 2004, 186 (17), 5876–5882.

This is the problem with acronyms; once established, it's best to stay consistent so as not to confuse the reader. Or simply use numbers.

Line 280 and elsewhere: Is it scientifically necessary to use 'cargo' in place of the term ligand or substrate?

Lines 285-285: "...positioning the pi electron system of an aromatic amino acid adjacent to the P-H atom of the ligand is a conserved molecular mechanism governing the specificity of the PBPs for reduced phosphate ligands..."

For context it is worthwhile to compare (just one line) the predicted phosphite binding mode by phosphite dehydrogenase PtxD, which uses Arg, His, and amide backbone interactions with the ligand (in this case, sulphite). Of course, in this case the P-H bond is predicted to point towards the aromatic nicotinamide ring of NADPH.

Zou, Y.; Zhang, H.; Brunzelle, J. S.; Johannes, T. W.; Woodyer, R.; Hung, J. E.; Nair, N.; van der Donk, W. A.; Zhao, H.; Nair, S. K. Crystal Structures of Phosphite Dehydrogenase Provide Insights Into Nicotinamide Cofactor Regeneration. *Biochemistry* 2012, 51 (21), 4263–4270.

Interestingly, sulfite is a competitive inhibitor of PtxD as a phosphite mimic (see above reference). Does sulfite bind to Te_PtxB?

Line 441: change to "...in the future."

Page 18, second paragraph of Discussion. I had a hard time with this paragraph, which tends to meander from the opening sentence. What are the authors trying to say? In part, the authors seem to be criticizing in vivo experiments which use P ligand concentrations that are well above the environmental values, which can be misleading in predicting the ligand specificity of the periplasmic binding protein. Fair enough.

However, I think the paper by Hirota et al is unfairly singled out for attention. For the purposes of Hirota et al, the concerns of the authors are largely irrelevant. Hirota et al clearly show that HtxBCDE is necessary to transport phosphite into a transporter deficient strain of E. coli, creating a phosphite dependent strain, which was the goal of their study. HtxB had sufficient affinity for phosphite to enable effective transport (ie: enabled survival of the cell). They did not need in vitro biochemical studies to achieve this goal; in fact, in vitro studies may have deterred them from using HtxBCDE. And since phosphite is cheap, using 1 mM for selection is not a problem. So why "would it be interesting to test the growth of engineered E. coli strains with lower P ligands in the future".? I note that Hirota et al had already done this to demonstrate that their engineered E. coli strain could not escape the lab to live on environmental levels of phosphite.

Figure 3, (e) one of the Tyr labels (lower right) is cut off.

Figure 6, Need numbers above the sequences in (b). Perhaps use PtxB numbering? Also, I had a hard time understanding the labels of each sequence. Just give the accession number of the protein or the trivial name (eg: Te_PtxD).

Line 338 and caption of Figure 6: 2-aminoethylphosphonate is misspelled

Reviewer #3 (Remarks to the Author):

The revised version of Hitchcock and colleagues addresses satisfactorily the major points raised by myself and the other reviewers, and most importantly presents the minimal mutational analysis and relative binding assays needed to support the proposed models on transporter-substrate interactions.

Point-by-point response to referees' comments:

Reviewer #1 (Remarks to the Author):

The authors have addressed all my queries. They have included additional functional data and much improved MST measurements including mutagenesis. Since the first review, they have determined the structure of an apo SBD. They have also presented convincing validation electron density maps for the substrates.

We are pleased that we have addressed all the queries of Reviewer 1 and thank them for their comments.

Only two minor comments:

1. I think the apo structure should be presented at the beginning of the results section than later. It will flow much better.

We have considered this suggestion, but we feel that if we present the apo-structure at the start of the results, we will need to move the comparison with the closed structure to a later section, splitting the information and potentially muddying the analysis of the closed structure. For this reason, and as the apo-structure is not the main focus of the paper, we think it would be better presented after the ligand bound PtxB/PhnD structures; we have therefore moved it up to before the section on HtxB. We hope this compromise satisfies the reviewer.

2. The authors have missed the opportunity to do some more in depth analysis after the anisotropic refinement of the Te_PtxB/MPn, Pm_Phnd/phosphite and Ps_PtxB/MPn structures. It would have been interesting to include a supplementary figure with plotting the ellipsoids for the ligands and discuss a bit further the selectivity.

Following the reviewer's suggestion, we have generated ellipsoids around the binding pocket for the three structures that are better than 1.5Å resolution (Ps_PtxB/MPn, Pm_Phnd/Pst and Te_PtxB/MPn) (see below). This has been added as supplementary figure 14. We have referred to the figure in the main text at lines 222-227, but the analysis is limited as we do not have comparable resolution structures of the same protein with the different ligands.

Reviewer #2 (Remarks to the Author):

The revised document by Bisson et al addresses the majority of my initial concerns, particularly with the MST binding assay and the mutagenesis experiments. I believe the paper is suitable for publication.

We are pleased that we have addressed the majority of the initial concerns of Reviewer 2, and the reviewer feels the paper is now suitable for publication.

The authors may want to consider the following comments.

We would like to thank the reviewer for these additional interesting comments and ideas and reply to each individually below. We have numbered the comments for clarity.

1. Line 91-92: "...because their utilization requires fewer protein components ...compared to phosphonate breakdown...."

I'm not sure if this is a valid comparison of phosphite and phosphonates as Pi sources.

CP-lyase is not the only system that is used by bacteria to catabolize organophosphonates; there are also the CP hydrolases (eg: phosphonate, phosphonoacetate hydrolase, etc) and the oxidative PhnY/PhnZ pathway. These are simple systems that do not require ATP and liberate Pi from phosphonates in 1-2 chemical steps; however, they are much more substrate specific relative to CP-lyase.

We agree with the reviewer that only taking into account the substrate promiscuous C-P lyase is not a fair comparison of phosphite/hypophosphite versus phosphonate utilisation as Pi sources given that there are other more substrate specific systems. Furthermore, such a comparison is now irrelevant to the reader in the context of our paper. As such we have deleted this section.

2. Line 100: "ligand specificity" mentioned as not being well studied.

We have removed "ligand specificity" and now only state that the mechanism of high-affinity ligand binding is not well studied.

3. Lines 157-158: "...suggests that....monoanionic phosphite is the predominantly bound form"

This builds off my earlier comment about the ionization state of the active site His. Could the ionization state be probed more directly using 31P-NMR spectroscopy of phosphite bound to PhnD? The one-bond J(PH) coupling constant is known to be different for dianionic (PHO²⁻) and monoanionic (PHO³⁻) forms:

Eykyn, T. R.; Kuchel, P. W. Scalar Couplings as pH Probes in Compartmentalized Biological Systems: 31P NMR of Phosphite. *Magn. Reson. Med.* 2003, 50 (4), 693–696.

And 31P-NMR has long been used to examine Pi and phosphonates bound to proteins:

Chlebowski, J. F.; Armitage, I. M.; Tusa, P. P.; Coleman, J. E. 31P NMR of Phosphate and Phosphonate Complexes of Metalloalkaline Phosphatases. *J. Biol. Chem.* 1976, 251 (4), 1207–1216.

I think this could be a relatively painless experiment (famous last words!) to apply to the best behaved (ie: most soluble, least prone to precipitation) PhnD variant. I will not insist upon this experiment for the paper, but it's worth considering. It might also be a way to probe the P-H aromatic pi system interaction.

Based on the structural evidence it is reasonable to suggest that the mono-anionic form of phosphite is bound, as discussed in the paper. We would like to thank the reviewer for their suggestions regarding the NMR. This is something that we are planning on doing, but as its unlikely to be painless or quick we feel that it is beyond the scope of the current paper.

4. Lines 182-183 and Table 3

The mutagenesis results are intriguing and not what I would have predicted! The Y208A substitution in Te_PtxD was less detrimental than the more conservative Y208F substitution for phosphite binding. If deleting the hydroxyl of tyrosine prevents closing of the cap, the alanine substitution would do this as well, plus lose any pi interaction with the phosphite P-H. Any thoughts?

We are glad the reviewer found the results of our mutations interesting. We have lots of thoughts, some more convincing than others, but all tending towards pure speculation. Perhaps without a hydrogen bond donor/acceptor (i.e. the -OH of the Tyr) the pi system of a Phe cannot be stabilised next to the ligand, therefore full domain closure cannot be achieved. Of course, the Phe may also be in a different position to the Tyr in the open structure, which may not be compatible with undergoing a conformational change that

allows it to form a functional cap in the closed complex. Or maybe the change in affinity is due to the charge distribution in the Phe, which is different to a Tyr. Considering the Y208A mutation, this may generate space above the binding pocket allowing different bridging interactions to form between the domains, which may achieve sub-optimal closure of the protein and hence some residual low-level binding. As ever with mutations, they rarely paint a full picture and so we felt analysis such as this was too speculative to be included in the paper, however, it is something that we would like to investigate further in future experiments. Nevertheless, these mutational analyses do show the importance of the tyrosine in constructing the binding pocket, which was the point of the mutagenesis and which we emphasise in lines 190-206.

5. “1 to 2 orders of magnitude weaker” The K_d value changes from 0.17 μM in the WT to 135 μM in Y208A (800 fold) and 295 μM in Y208F (1735 fold). This is more than a 10 or 100 fold change.
We thank the reviewer for spotting this error, which has now been amended.

6. Should also fix up the significant figures of the K_d values in Table 3 (eg: 51.3+/-1.76 should be 51+/-2). We are a little confused. We are happy to report all values to 2 significant figures, but the data of high enough quality to report to a higher degree of accuracy. If this can this be clarified, we can adjust the values as necessary in the text and in Tables 1 and 3.

7. Table 3 abbreviations (and elsewhere): I haven’t seen Pst and HPst used as acronyms for phosphite and hypophosphite, respectively. Pst has been used previously in the literature, most prominently by Wanner, for the bacterial Phosphate Specific Transport system.

Hsieh, Y.-J.; Wanner, B. L. Global Regulation by the Seven-Component Pi Signaling System. *Curr Opin Microbiol* 2010, 13 (2), 198–203.

However, Pt and HPt have been used previously for phosphite and hypophosphite. See:

White, A. K.; Metcalf, W. W. The Htx and Ptx Operons of *Pseudomonas Stutzeri* WM88 Are New Members of the Pho Regulon. *J. Bacteriol.* 2004, 186 (17), 5876–5882.

This is the problem with acronyms; once established, it’s best to stay consistent so as not to confuse the reader. Or simply use numbers.

To most of the scientific community Pt is platinum, so we tried to find an acronym that would be less conflicting, but we agree that Pst and HPst are not ideal. We have therefore removed the acronyms for phosphite and hypophosphite from the manuscript using the full names in the text, figures and tables.

8. Line 280 and elsewhere: Is it scientifically necessary to use ‘cargo’ in place of the term ligand or substrate? No, it is not scientifically necessary. All uses of ‘cargo’ have been replaced with ligand.

9. Lines 285-285: “...positioning the pi electron system of an aromatic amino acid adjacent to the P-H atom of the ligand is a conserved molecular mechanism governing the specificity of the PBPs for reduced phosphate ligands...”.

For context it is worthwhile to compare (just one line) the predicted phosphite binding mode by phosphite dehydrogenase PtxD, which uses Arg, His, and amide backbone interactions with the ligand (in this case, sulphite). Of course, in this case the P-H bond is predicted to point towards the aromatic nicotinamide ring of NADPH.

The predicted phosphite binding site in PtxD is quite different to that of the PtxB/PhnD proteins presented here, being much more like that of typical dehydrogenase enzymes from this particular family. PtxD uses a series of hydrogen bonding interactions to the amide backbone, an Arg and a His to form the phosphite binding site, rather than the cluster of predominantly Ser/Thr and Tyr residues, as observed in PtxB/PhnD. We are not sure that the PtxD structure is relevant to this paper and adding in this comparison would, we feel, muddy the waters. In PtxD the enzyme has been designed to bind a phosphite substrate in such a way that facilitates hydride transfer to NADP(+). Thus it is not surprising (or particularly relevant) that the active site of the PtxD enzyme is constructed (with the P-H bond of phosphite pointing towards the C4 of NADP(+)), in a different way to the ligand binding pockets of the phosphite PBPs described here.

Zou, Y.; Zhang, H.; Brunzelle, J. S.; Johannes, T. W.; Woodyer, R.; Hung, J. E.; Nair, N.; van der Donk, W. A.; Zhao, H.; Nair, S. K. Crystal Structures of Phosphite Dehydrogenase Provide Insights Into Nicotinamide Cofactor Regeneration. *Biochemistry* 2012, 51 (21), 4263–4270.

10. Interestingly, sulfite is a competitive inhibitor of PtxD as a phosphite mimic (see above reference). Does sulfite bind to Te_PtxB?

The way phosphite is predicted to bind to the phosphite dehydrogenase enzyme, PtxD, is very different to how phosphite is coordinated by the PBPs in our study (as discussed in response to the previous comment). Importantly, sulphite binding by PtxD requires that the NAD cofactor (Relyea et al. 2005 *Biochemistry* **44**, 6640-6649) and whilst sulphite has a trigonal pyramidal shape like phosphite, it has a lone pair rather than a proton at the R1 position, and thus is not equivalent to the P-H bond in phosphite, which is a selective factor in binding to our family of PBPs. We did try to measure binding affinity for sulphite with Te_PtxB via our MST assay, but unfortunately the protein precipitated in the presence of sodium sulphite. While we agree with the reviewer that it would be interesting to look at whether sulphite and other anions with similar geometry to phosphite (e.g. arsenite) can competitively bind to the proteins investigated in our study, as we explained to Reviewer #1 in our previous response, this would be a considerable amount of additional work and we feel such a significant undertaking is for a future study and is not required for our current paper.

11. Line 441: change to “...in the future.”

This section has been changed (see response to comment 12 below).

12a. Page 18, second paragraph of Discussion. I had a hard time with this paragraph, which tends to meander from the opening sentence. What are the authors trying to say? In part, the authors seem to be criticizing *in vivo* experiments which use P ligand concentrations that are well above the environmental values, which can be misleading in predicting the ligand specificity of the periplasmic binding protein. Fair enough.

On reflection we agree with the reviewer that this part of the discussion was unclear. We would like to point out that we were not meaning to criticize the *in vivo* experiments of others, but as the reviewer suggests, we wanted to get across that the use of concentrations above what are environmentally relevant mean interpretation of the ‘real’ ligand specificity from such experiments can be misleading. Indeed, as the reviewer points out below (see 12b), Hirota *et al.* may have ruled out HtxBCDE for their study based solely on our data and so the same argument could be made by the authors of *in vivo* studies about *in vitro* work, hence why the two studies are complementary to one another. Accordingly, this section has been significantly modified and, we hope, is now much improved. We thank the reviewer for helping us make the discussion clearer and making sure we did not unintentionally say something which sounded like we were critical of others work.

12b. However, I think the paper by Hirota et al is unfairly singled out for attention. For the purposes of Hirota et al, the concerns of the authors are largely irrelevant. Hirota et al clearly show that HtxBCDE is necessary to transport phosphite into a transporter deficient strain of *E. coli*, creating a phosphite dependent strain, which was the goal of their study. HtxB had sufficient affinity for phosphite to enable effective transport (ie: enabled survival of the cell). They did not need *in vitro* biochemical studies to achieve this goal; in fact, *in vitro* studies may have deterred them from using HtxBCDE. And since phosphite is cheap, using 1 mM for selection is not a problem.

By no means did we mean to single out the work of Hirota and colleagues for any criticism, conversely we wanted to highlight an interesting and very recent study that was published while our paper was in revision and is largely complementary to our work. The point we were trying to get across is that phosphate uptake by the PtxABC containing strain can be explained by our micromolar K_d of PtxB/phosphate. However, phosphite uptake by HtxBCDE is not consistent with our binding data (we could not detect binding of phosphite by HtxB in our assays), but as the reviewer points out, it works *in vivo* and supplying phosphite at 1 mM is not a problem, so for the biocontainment strategy of Hirota and colleagues, the K_d doesn't matter! We wanted to try and explain to the reader why this discrepancy might occur. We have tried to do this but also make it clear that we are not critical of the Hirota *et al.* paper and that how HtxB can bind phosphite, which based on our structure with hypophosphite must involve re-configuration of the binding pocket to make room for the extra oxygen atom of phosphite, requires further structural investigation in the future.

12c. So why “would it be interesting to test the growth of engineered *E. coli* strains with lower P ligands in the future”.? I note that Hirota et al had already done this to demonstrate that their engineered *E. coli* strain could not escape the lab to live on environmental levels of phosphite.

I think this point was badly made in the previous version of the discussion and it has been mostly removed, however we feel we should explain our thinking to the reviewer. What we were trying to get across is that the HtxBCDE strain grows with 1 mM phosphite (way above environmentally relevant concentrations) but not at 5 μ M phosphite (more like an environmentally relevant concentration, but still higher than the maximum recorded environmental concentration), as the reviewer says; we still include a statement to such effect in the discussion. What we thought would additionally be interesting is to explore if the PtxABC strain was grown on lower concentrations: (1) at what (if any) concentration does phosphate (K_d = approx. 50 μ M) no longer supports growth but phosphite (K_d = approx. 170 nM) still does, and (2) does 5 μ M phosphite support growth of this strain, which has a dedicated high affinity phosphite transporter? In a similar vein, in a strain with HtxA, PtxD and PtxABC what concentration of hypophosphite (PtxB K_d for hypophosphite = approx. 4 μ M so in between phosphate and phosphite) is permissive for growth. Similarly, with the HtxBCDE transporter, what is the cut off concentration between 5-1000 μ M for phosphite supporting growth, and is this concentration lower for hypophosphite? We note that the 2-oxoglutarate requirement of HtxA may make some of these comparisons difficult, however overall we think this would yield some very interesting data.

We also note that Hirota et al discussed that the structural basis of substrate recognition by HtxB was unknown and would be useful for their work (in working out how many mutations would be required for HtxB to transport phosphate). We hope that they will find our work interesting and useful, as we did theirs.

Again, we thank the reviewer for their comments and we hope they find the modified discussion clearer.

13. Figure 3, (e) one of the Tyr labels (lower right) is cut off.

We thank the reviewer for spotting this error and have sorted the problem (note this is now Figure 2).

14. Figure 6, Need numbers above the sequences in (b). Perhaps use PtxB numbering? Also, I had a hard time understanding the labels of each sequence. Just give the accession number of the protein or the trivial name (eg: Te_PtxD).

Please note this is now Figure 5. We agree that panel b should have numbers and the sequence labels were unclear. The sequences have been re-named to make it clearer what each one is.

For the numbering, we debated for some time about how to make this clear. In the end we decided to number the position in the alignment above the sequences, and the residue numbers have been added for each of the protein sequences at the end of each row. This has been done in two ways, the first (in red/orange) such that the residue numbering is in keeping with the same numbering used for describing the structures (all proteins were expressed without their N-terminal periplasmic targeting peptide, so just numbering from Met1 would confuse the reader, therefore we used either the numbering from the structure, where applicable, or the predicted first residue following the predicted signal peptide cleavage site for other proteins). Secondly, (in black parentheses) we give the residue number of the protein including the signal peptide. We feel this will give the reader all the information they would need when analysing the sequence alignments. The legend has been altered to reflect this change.

15. Line 338 and caption of Figure 6: 2-aminoethylphosphonate is misspelled
We have corrected this error and carefully checked the spelling throughout.

Reviewer #3 (Remarks to the Author):

The revised version of Hitchcock and colleagues addresses satisfactorily the major points raised by myself and the other reviewers, and most importantly presents the minimal mutational analysis and relative binding assays needed to support the proposed models on transporter-substrate interactions.

We are pleased that we have satisfactorily addressed the major points raised by Reviewer 3 and thank them for their comments.